# Q586B2 is a crucial virulence factor during the early stages of *Trypanosoma brucei* infection that is conserved amongst trypanosomatids

Benoit Stijlemans [1,2] ✉, Patrick De Baetselier[1,2], Inge Van Molle[3,4], Laurence Lecordier[5], Erika Hendrickx[6], Ema Romão[7], Cécile Vincke[1,2], Wendy Baetens[1,2], Steve Schoonooghe[7], Gholamreza Hassanzadeh-Ghassabeh[7], Hannelie Korf[8], Marie Wallays[8], Joar E. Pinto Torres[1], David Perez-Morga[6,9], Lea Brys[1,2], Oscar Campetella[10], María S. Leguizamón[10], Mathieu Claes [11], Sarah Hendrickx[11], Dorien Mabille [11], Guy Caljon [11], Han Remaut [3,4], Kim Roelants[12], Stefan Magez[1,13], Jo A. Van Ginderachter [1,2,14] & Carl De Trez [1,14]

Human African trypanosomiasis or sleeping sickness, caused by the protozoan parasite *Trypanosoma brucei*, is characterized by the manipulation of the host's immune response to ensure parasite invasion and persistence. Uncovering key molecules that support parasite establishment is a prerequisite to interfere with this process. We identified Q586B2 as a *T. brucei* protein that induces IL-10 in myeloid cells, which promotes parasite infection invasiveness. Q586B2 is expressed during all *T. brucei* life stages and is conserved in all Trypanosomatidae. Deleting the Q586B2-encoding *Tb927.6.4140* gene in *T. brucei* results in a decreased peak parasitemia and prolonged survival, without affecting parasite fitness in vitro, yet promoting short stumpy differentiation in vivo. Accordingly, neutralization of Q586B2 with newly generated nanobodies could hamper myeloid-derived IL-10 production and reduce parasitemia. In addition, immunization with Q586B2 delays mortality upon a challenge with various trypanosomes, including *Trypanosoma cruzi*. Collectively, we uncovered a conserved protein playing an important regulatory role in Trypanosomatid infection establishment.

African trypanosomes are extracellular flagellated protozoan parasites with a complex digenetic life cycle that can cause debilitating diseases of both medical and veterinary importance in sub-Saharan Africa[1,2]. Upon transmission through the bite of their blood-feeding vector (*i.e.*, the tsetse fly, *Glossina spp.*), these parasites can cause fatal diseases in mammals, commonly called sleeping sickness in humans (Human African Trypanosomosis, HAT) or Nagana (Animal African Trypanosomosis, AAT) in domestic livestock. HAT is caused by either *Trypanosoma brucei gambiense* (accounting for over 95–97% of the cases) or *Trypanosoma brucei rhodesiense* (accounting for the remainder of the cases)[1,3]. HAT is classified as a neglected tropical disease by the World Health Organization (WHO)[4–6], exhibiting high morbidity and mortality rates, affecting millions of impoverished populations in the developing world, displaying a

limited response to chemotherapy, and showing unresponsiveness towards vaccination.

Nagana (*Trypanosoma brucei brucei*, *Trypanosoma congolense*, *Trypanosoma vivax*), Surra (*Trypanosoma evansi*) or Dourine (*Trypanosoma equiperdum*) are the major forms of AAT[2]. Since the occurrence of Nagana is an important constraint for cattle production, it has a great impact on the nutrition of millions of people living in AAT endemic areas, and on the agriculture economics, resulting in an estimated annual economic cost of about US\$ 4 billion[7,8]. Murine models, which are more easily amenable compared to cattle or other domestic animals, are very useful tools to study AT. In this respect, a murine *T. b. brucei* infection exhibits characteristics of HAT and is often the model of choice.

Due to millions of years of co-evolution, trypanosomes have been able to thwart host innate immune responses and escape early immune recognition, allowing the initiation of infection and a sufficiently long survival in their mammalian host to complete their life cycle and transmission[9,10]. However, the molecular machinery that *Trypanosoma* species employ to manipulate host immune responses is largely unknown. In other protozoan parasites, including *Plasmodium*, *Entamoeba*, *Toxoplasma*, and *Leishmania*, a homolog of the mammalian Macrophage Migration Inhibitory Factor (MIF) was identified, which functions as a virulence factor aiding in the establishment or persistence of infection by modulating the host innate immune response[11,12]. Though the sequence identities between mammalian and protozoan MIF only range between 20 and 27%, they share a well-conserved trimeric architecture[12–14]. However, whether trypanosomes also contain a MIF-like molecule that facilitates the infection was an outstanding question.

Here, we show that trypanosomes, despite their close relationship to *Leishmania*, did not harbor a MIF homolog. Instead, we identified Q586B2, a protein that is localized in vesicles and harbors tautomerase activity (an isomerase enzyme catalyzing the interconversion of keto- and enol-groups). A constitutive or inducible deficiency of the *Tb927.6.4140* gene, coding for Q586B2, does not affect parasite fitness as such, but decreases parasite establishment and prolongs survival of the host, which is mechanistically coupled to its potent IL-10-inducing capacity in myeloid cells. In addition, Q586B2 deficiency increases the propensity of parasites to differentiate into short stumpy forms. We provide evidence that blocking this protein or using it in a vaccination setting holds great promise to hamper trypanosome-mediated diseases.

## Results

### In silico analysis identified Q586B2 as a novel *T. brucei* protein exhibiting structural homology with the *T. cruzi* protein Q4D6Q6

In an attempt to identify a MIF homolog within AT, the *Leishmania* MIF protein sequence (Q4Q413) was blasted against the available trypanosome genome (*Trypanosoma brucei brucei* TREU 927) within the Wellcome Trust Sanger Institute database (http://www.sanger.ac.uk/resources/databases). However, no MIF homolog was detected in trypanosomes. We then screened for proteins with MIF-like characteristics, such as an amino-terminal proline residue immediately following the initial methionine residue and a maximum size of 115 amino acids, by using the first 25 amino acids of the Q4Q413 protein for blasting. This approach identified *Tb927.6.4140*, a gene located on chromosome 6, coding for a hypothetical unknown protein (Q586B2) with an expected size of 13 kDa (348 bp, 115 aa), and little sequence similarity with MIF. In accordance, the Protein Homology/analogy Recognition Engine Version 2.0 (Phyre²) (http://www.sbg.bio.ic.ac.uk/phyre2/html/page.cgi?id=index), that also analyzes structural homologies between proteins, yielded no homology between Q586B2 and MIF. However, Phyre² predicted a weak probability of homology (16% sequence identity and 51.5% structural homology confidence for the

first part of the protein, ranging from amino acid 13–70) with 4-oxalocrotonate tautomerase (4-OT) of the Archean species *Archaeoglobus fulgidus* (Fig. 1a, upper panel). Phyre² also uncovered a high homology to Q4D6Q6[15], a conserved kinetoplastid-specific protein from *Trypanosoma cruzi* with unknown function (55% sequence identity and 100% structural homology confidence) (Fig. 1a, lower panel). Using ColabFold[16,17], the monomeric structure of Q586B2 was predicted and compared to the known structure of monomeric Q4D6Q6 (Fig. 1b(i)). In this algorithm, each residue receives a local confidence measure on a scale from 0 to 100 (i.e., the pLDDT or predicted local distance difference test score value; 100 representing the highest confidence). The prediction of the Q586B2 protein structure shows pLDDT values higher than 90 for 86 out of 115 amino acid residues (Fig. 1(i), (ii)). Loop regions show a slightly lower confidence, but the pLDDT is still higher than 70. The level of confidence is only low (pLDDT between 50 and 70) for residues 42–46, which are part of loop 3 between α1 and β3 (highlighted in yellow). Regions with a low pLDDT may be unstructured, either intrinsically disordered or structured only in the context of a larger complex. As shown in Fig. 1b(i), the Q586B2 monomer has a high structural homology with monomeric Q4D6Q6[15], for which the crystal structure (PDB code 6xyb) showed loop 3 to be unstructured.

As Q4D6Q6 reportedly forms a homo-tetramer[15], we used ColabFold to predict the likelihood of Q586B2 to organize into a similar multimeric structure (Fig. 1c). The Predicted Aligned Error (PAE) plot shows that ColabFold is confident about the relative domain positions (Fig. 1c, left panel). The color at (x, y) indicates the expected position error at residue x if the predicted and true structures were aligned on residue y. The low PAE (colored blue) along the diagonal indicates a correct prediction of the monomeric structures (A–D), whereas a low error along the different axes of the PAE plot indicates a correct relative orientation of the monomers in the oligomeric structure (A–D). As shown from the overlay with the Q4D6Q4 tetrameric structure (Fig. 1c, right panel), Q586B2 is predicted to form a very similar propeller-like homo-tetramer, consisting of 4 monomers that adopt a ββαββαββ topology[15].

Collectively, these data demonstrate that trypanosomes do not harbor a bona fide MIF homolog, but uncovered Q586B2 as a protein with structural homology to the *T. cruzi* protein Q4D6Q6.

### Q586B2 is an evolutionary conserved protein within the related families Trypanosomatidae and Bodonidae

To chart the taxonomic distribution of proteins homologous to Q586B2 and Q4D6Q6, thereby elucidating the evolutionary origin of the underlying gene family, we used tBLASTn searches to screen a wide range of eukaryote genomes in the NCBI and TriTrypDB databases. These searches uncovered the presence of three paralogous genes (*Tb927.6.4140*, *Tb927.2.2770*, and *Tb10.26.0680*) in *T. b. brucei* as well as in the (sub)species *T. b. gambiense*, *T. congolense*, *T. equiperdum*, *T. evansi*, and *T. vivax*. A fourth paralogous gene was discovered in four other *Trypanosoma* species (*T. cruzi*, *T. rangeli*, *T. conorhini*, *T. Theileri*), as well as in all investigated species of the trypanosomatid genera *Leishmania*, *Angomonas*, *Crithidia*, *Endotrypanum*, *Herpetomonas*, *Leptomonas*, *Lotmaria*, *Paratrypanosoma*, *Phytomonas* and *Strigomonas*, and in *Bodo saltans*, a free-living (non-parasitic) species of the closely related family Bodonidae[18] (Table S1). Remarkably, BLAST searches did not reveal any related genes in aquatic *Trypanosoma* species. In addition, no genes homologous to *Tb927.6.4140* (encoding Q586B2) were found in any kinetoplastid genome outside Bodonidae and Trypanosomatidae, nor in those of any other eukaryote supergroup. This pattern of occurrence suggests that the gene family to which *Tb927.6.4140* belongs arose in a direct ancestor of the Bodonidae and Trypanosomatidae sister families. Comparative alignment shows that the different proteins encoded by the closest related genes found in other trypanosomatid and bodonid taxa share considerable

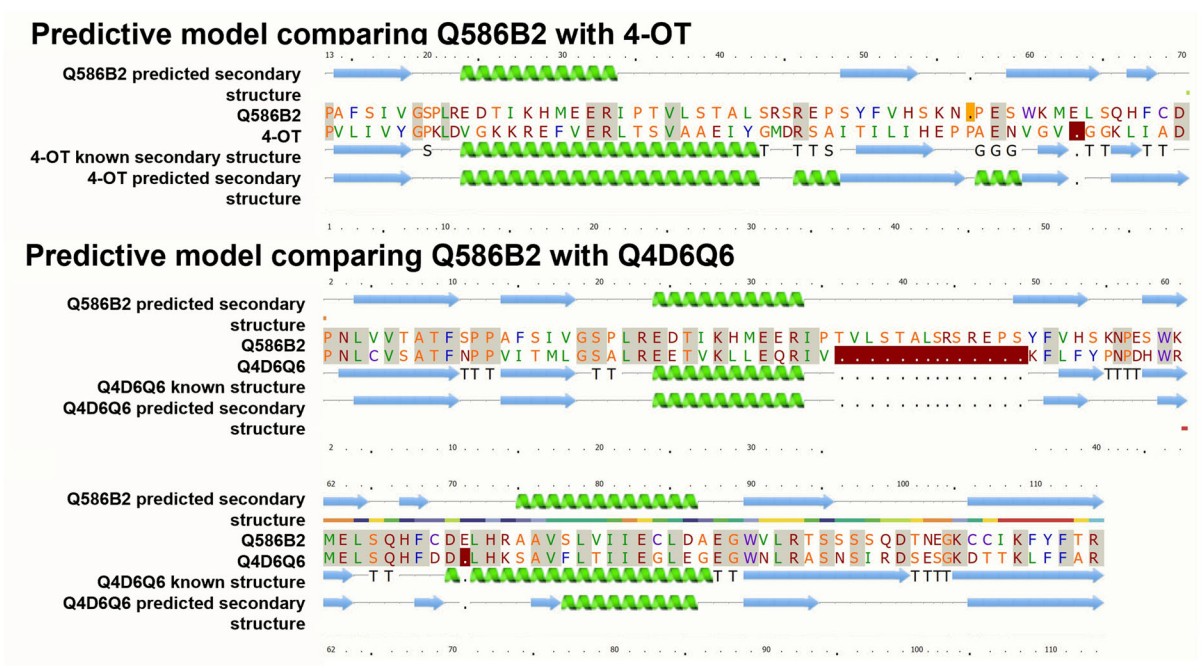

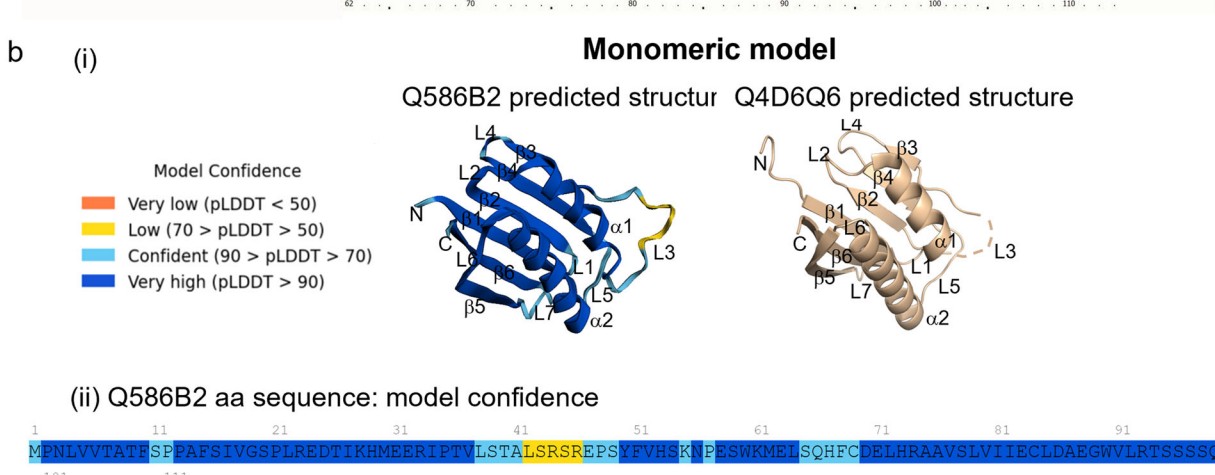

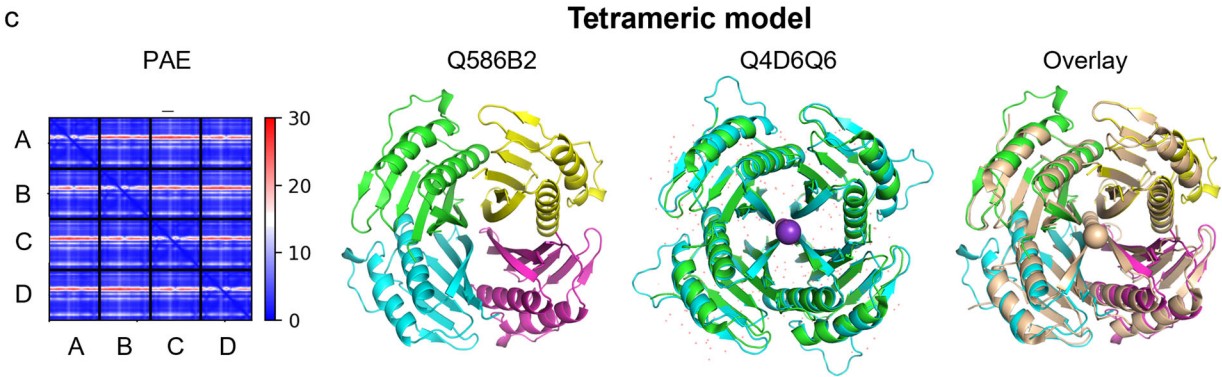

sequence similarity to Q4D6Q6 and Q586B2 (Fig. S1a). Moreover, Phyre² analyses predict with high confidence that the majority of them share the same secondary structure and folding pattern as Q4D6Q6, which for all proteins was selected by Phyre² as the folding template.

To investigate the evolutionary history of the genes encoding Q586B2 and its homologs, we conducted a Bayesian phylogenetic analysis based on an alignment of 126 gene sequences retrieved from the abovementioned taxa (Fig. S2). The resulting tree is composed of four major gene clades, each including one of the four paralogues found in most species. This pattern indicates that a single ancestral gene underwent a series of duplication events prior to the divergence of Bodonidae and Trypanosomatidae, yielding a gene family of four paralogues that remained stable in most taxa. The presence of only

**Fig. 1 | In silico identification of Q586B2 as a novel *T. brucei* protein exhibiting structural homology with the *T. cruzi* protein Q4D6Q6. a** Upper panel: The *Leishmania* MIF protein sequence (Q4Q413) was blasted against the available trypanosome genome and identified Q586B2 as a potential candidate. Phyre² (Protein Homology/analogy Recognition Engine V 2.0) analysis predicts a 16% sequence identity and 51.5% confidence of structural homology with 4-OT (4-oxalocrotonate tautomerase, c3m20A) for the first half of the Q586B2 protein. Residues with 100% homology (highlighted in gray), alpha helices (green), and beta strands (blue). Lower panel: Phyre² analysis predicts a high homology of Q586B2 to Q4D6Q6 with 55% sequence identity and 100% structural homology confidence. Green helices represent α-helices, blue arrows indicate β-strands and faint lines indicate coils. The 'SS confidence' line indicates the confidence in the prediction (red: high confidence, blue: low confidence). **b** Upper panel: ColabFold predictions (https://www.alphafold.ebi.ac.uk) of the structure of Q586B2 (left) as a monomer, based on the Per-residue confidence (pLDDT: *i.e.*, a predicted local distance difference test

score), which is a per-residue measure of local confidence on a scale from 0 to 100. Regions with high (dark blue) and low pLDDT (yellow, these may be unstructured; either intrinsically disordered or structured only in the context of a larger complex). The crystal structure of the Q4D6Q6 monomer is shown on the right (pdb code 6xyb). **c** Left panel: Predicted Aligned Error (PAE) plot. The color at (x, y) indicates the expected position error at residue x if the predicted and true structures were aligned on residue y. A low PAE (colored blue) along the diagonal indicates correct prediction of the monomer structures, whereas a low error along the different axes of the PAE plot indicated correct relative orientation of the monomers in the oligomeric structure. A–D: the different monomers represented as consecutive residue numbers. Right panels: the ColabFold prediction of the Q586B2 structure and the crystal structure of Q4D6Q6 show a similar propeller-like tetramer consisting of 4 monomers that adopt a ββαββαββ topology, and overlay of the documented tetrameric structure of Q4D6Q6[15] with the predicted tetrameric structure of Q586B2 (pink).

three paralogues in *T. b. brucei* and closely related (sub)species is explained by a loss of one of the four paralogues within *Trypanosoma*. The genes encoding Q4D6Q6 (*TcCLB.511733.90*) and Q586B2 (*Tb927.6.4140*) are found deeply nested in one of these four gene clades (Fig. S2), explaining the presence of orthologous genes in all analyzed trypanosomatid genera as well as in *Bodo saltans*. The presence of orthologous genes in a broad taxonomic range of Trypanosomatidae as well as in their sister family Bodonidae implies that the ancestral gene of *Tb927.6.4140* arose before the onset of a parasitic lifestyle and long before the independent evolution of dixenous life histories in multiple trypanosomatid lineages (Fig. S1b).

### The Q586B2 protein is present within intracellular vesicles and at different life cycle stages, being conserved among trypanosomatids of the Kinetoplastida phylum

To obtain functional insights in the Q586B2 protein, we produced the protein recombinantly in *E. coli* WK6 cells by codon optimizing the *Tb927.6.4140* gene and cloning it with an N-terminal His₆- and FLAG-tag, allowing the removal of the tag after purification. Upon Q586B2 purification via IMAC and size exclusion chromatography, a uniform peak of approximately 55 kDa was observed, suggesting that the protein forms a tetramer (Fig. S3a). Since part of the Q586B2 protein showed resemblance to 4-OT (Fig. 1a), which belongs to the tautomerase superfamily, we first assessed its putative tautomerase activity. Q586B2 indeed exhibited a moderate tautomerase activity, which is significantly lower than the prototypical tautomerase activity of MIF (Fig. S3b). Moreover, in contrast to MIF, the Q586B2 tautomerase activity was not inhibited by the small molecule ISO-1, suggesting structural differences in the active site (Fig. S3b). This is in line with the fact that the predicted crystal structure of Q586B2 is a tetramer (see Fig. 1c), while MIF is a trimer.

To create novel research tools, we used the recombinant Q586B2 protein for the generation of nanobodies (Nbs), which are the antigen-recognition domains of camelid heavy chain-only antibodies. Following immunization of a llama and screening of its Nb repertoire for specificity against the recombinant Q586B2 protein, 8 distinct Nbs belonging to 8 different families based on the difference in their CDR3 regions (Fig. S4a(i), highlighted in yellow) could be retrieved, and after purification found to have nM affinity for binding to Q586B2 (Fig. S4a(ii)). Using flow cytometry, these Nbs were shown to recognize their target only on fixed and permeabilized parasites, suggesting that Q586B2 is localized intracellularly (Fig. S4b, left panels). Nb39 consistently gave the highest Q586B2 detection signal (Fig. S4b, right panel) and will be used throughout this manuscript, while Nb27 was unable to detect native Q586B2. Immunofluorescence microscopy using Nb39 confirmed that Q586B2 was localized in vesicles dispersed throughout the whole parasite (Fig. 2a). More specifically, Q586B2 partially co-localized with the glycosomal glycerol kinase (GK) and the vacuolar iron transporter

(VIT1) present in acidocalcisomes, with RAB5B and RAB11 present in early and recycling endosomes, respectively, and with RAB2B as a marker of the early exocytosis compartment (Fig. 2b). No co-localization with the lysosomal marker p67 nor endoplasmic reticulum marker BiP was observed.

Interestingly, Q586B2 was not only expressed in the bloodstream stage (Figs. S4b and 2a, b), but also in the procyclic (*i.e.*, early forms present in the fly midgut following a bloodmeal) and metacyclic (*i.e.*, forms present in the salivary glands of the fly preadapted for infection into the mammalian host; purified from *Glossina morsitans morsitans* salivary glands) stages of *T. b. brucei* (Fig. 2c(i)), indicating that it is constitutively expressed throughout the entire parasites' life cycle. Besides *T. b. brucei*, both the human pathogens *T. b. gambiense* and *T. b. rhodesiense* harbor this protein, as well as *T. evansi*, *T. congolense* and *T. vivax* (Fig. 2c(ii)), indicating that it is highly conserved among trypanosomes. These data also demonstrate that Nb39 recognizes a widely conserved epitope within the Q586B2 protein. Other trypanosomatids belonging to the phylum Kinetoplastida were also suggested to harbor a paralogous gene (Fig. S2). Using Nb39, we demonstrated the expression of a homologous protein in *T. cruzi*, *Leishmania major*, and *L. infantum* species (Fig. 2c(iii)), albeit at a lower level than in trypanosomes (Fig. 2c, right panel). Conversely, Q586B2 was not present in the parasitic protist *P. falciparum* belonging to the phylum Apicomplexa.

Together, Q586B2 was shown to be present in most trypanosomatids belonging to the phylum Kinetoplastida, being localized in intracellular vesicles and expressed throughout the *T. b. brucei* parasite's life cycle.

### *Tb927.6.4140*-deficiency attenuates first-peak parasitemia, leading to a prolonged survival of the host

To study the role of Q586B2 in AT biology, we generated *Tb927.6.4140*-knock out (*Tb927.6.4140*-KO) *T. b. brucei* parasites using the 90:13 pleiomorphic strain and a similar approach as described before[19]. PCR confirmed the successful deletion of the gene (Fig. S5a). *Tb927.6.4140*-KO parasites exhibited a similar in vitro proliferation rate as compared to WT parasites, indicating that the absence of Q586B2 does not affect the parasite's fitness (Fig. 3a). To determine whether *Tb927.6.4140*-deficiency affects the parasite's presence within the tsetse fly vector, we investigated midgut infection establishment in *Glossina morsitans morsitans* flies following feeding on WT or *Tb927.6.4140*-KO trypanosome-infected blood. The number of tsetse flies exhibiting a midgut infection, as well as the midgut parasite load, was similar in both conditions (Fig. 3b, left and right panel). Together, these data suggest that Q586B2 is dispensable for parasite fitness in vitro and survival within the tsetse fly vector.

To study whether Q586B2 affects parasitemia in the murine host, C57BL/6 mice were intraperitoneally injected with 5000 WT or *Tb927.6.4140*-KO *T. b. brucei* parasites, and infection parameters were

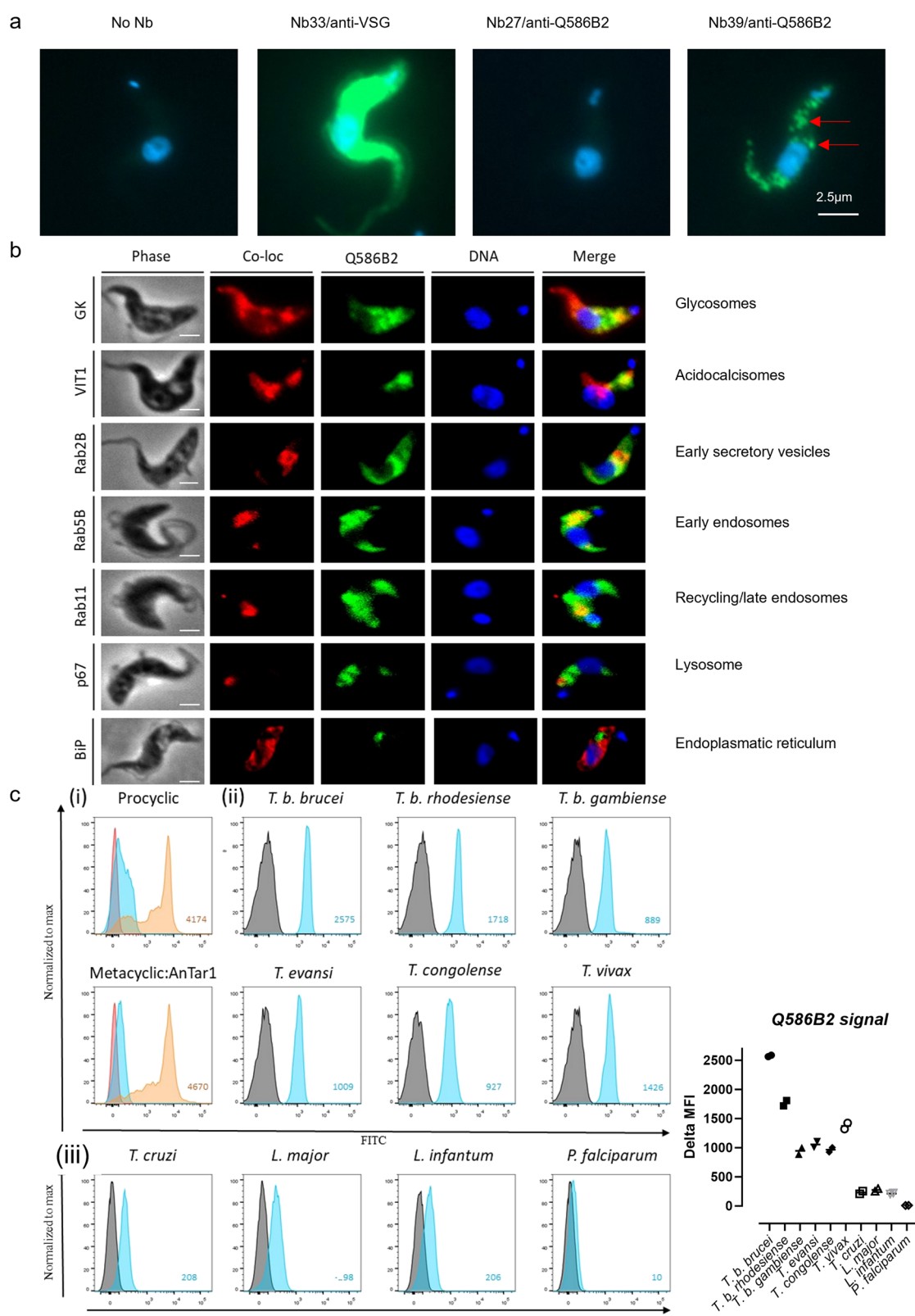

monitored. *Tb927.6.4140*-KO *T. b. brucei*-infected mice exhibited a significantly lower first-peak parasitemia (~80% reduction) and a significantly prolonged survival time (median survival time WT-infected mice: 22 ± 4 days vs. *Tb927.6.4140*-KO-infected mice: 56 ± 20 days) (Fig. 3c, d). The prolonged survival observed in *Tb927.6.4140*-deficient parasite-infected mice correlated with a significantly reduced AT-associated immunopathology. Indeed, compared to WT parasite-

infected mice, *Tb927.6.4140*-deficient parasite-infected mice exhibited reduced anemia as well as serum ALT (liver-specific injury marker) and AST (more general tissue injury marker) levels at day 6 p.i. (acute phase) and day 22 p.i. (chronic phase), suggesting a lower immunopathology (Fig. S6). A possible explanation could be that due to the lower parasite levels observed in the *Tb927.6.4140*-deficient parasite-infected mice at the early stages of infection, less parasite-derived

**Fig. 2 | Q586B2 is expressed in intracellular vesicles throughout different infection stages and in various trypanosomatid species. a** Immunofluorescence microscopy showing representative images of purified, fixed, and permeabilized *T. b. brucei* (AnTat1.1E) parasites stained with dapi (i.e., nucleus) alone or in presence of anti-Q586B2 Nbs (Nb27 and Nb39) or an anti-VSG Nb (Nb33). **b** Co-localization experiments using fixed V5-tagged *T. b. brucei* (90:13 strain) parasites stained with different primary antibodies (1:100 α-V5 IgG, 1:1000 α-p67 IgG, 1:1000 α-BiP IgG and 1:50 Nb39), and after washing with fluorochrome-conjugated secondary antibodies. After washing, samples were mounted with ProLong Gold Antifade with DAPI mounting medium. The images were taken using a Zeiss Axio Imager M2 wide field (scale bar: 2.5 μm). Results of 2 independent experiments performed in triplicate are shown in (**a**) and (**b**). **c** Representative flow cytometric anti-Q586B2 Nb binding profile using Nb39 on different purified, fixed, and permeabilized parasites (Gating and conditions as described in Fig. S4b). (i) Parasites used are *T. b. brucei* bloodstream forms (AnTat1.1E), procyclic forms and metacyclic forms. Signals of parasites without staining (red), parasites in presence of Alexa-488 labeled anti-HA IgG alone (blue), and parasites in presence of Nb39 and Alexa-488 labeled anti-HA IgG (orange) are shown. (ii) Parasites used are *T. b. brucei*, *T. b. gambiense*, *T. b. rhodesiense*, *T. evansi*, *T. congolense*, *T. vivax*, *T. cruzi*, *L. major*, *L. infantum* and *Plasmodium falciparum*. Signals of parasites in presence of Alexa-488 labeled anti-HA IgG alone was used as negative control (black/gray). **c** (right panel) Representative dot plot of 3 independent experiments showing the median fluorescence intensity (MFI) of anti-Q586B2 Nb39 following binding to fixed and permeabilized parasites. Each condition consists of technical duplicates (*n* = 2).

PAMPs (*i.e.*, VSG, CpG,…) are released that in turn will trigger a reduced inflammatory immune response in infected mice[20]. To delineate the exact time window within which Q586B2 functionality is important, we generated doxycycline-inducible *Tb927.6.4140*-knockdown (*Tb927.6.4140*-KD) *T. b. brucei* parasites using the 90:13 pleiomorphic strain. In vitro culture of inducible *Tb927.6.4140*-KD parasites in the presence of 1 μg/ml doxycycline is causing a full knockdown of the Q586B2 protein within 48 h of doxycycline exposure (Fig. S5b). Based on these data, WT or *Tb927.6.4140*-KD parasites were cultured for 48 h in presence of doxycycline to ensure full knockdown of the Q586B2 protein upon inoculation and subsequently injected into C57BL/6 WT mice that were also administered doxycycline in the drinking water already 2 days prior to infection to ensure a systemic presence of the antibiotic at the start of infection. As an internal control for each parasite strain, mice were given drinking water without antibiotic. While doxycycline administration did not affect the first-peak parasitemia in WT parasite-infected mice, it significantly lowered this infection parameter in mice infected with *Tb927.6.4140*-KD parasites (Fig. 3e), phenocopying infections with full *Tb927.6.4140*-KO parasites (Fig. 3c). Along the same line, the inducible *Tb927.6.440*-KD-infected mice survived significantly longer than the WT control group (WT: 24 ± 3 days, *Tb927.6.4140*-KD: 34 ± 5 days) (Fig. 3f). Of note, mice infected with *Tb927.6.4140*-KD parasites that received normal drinking water showed a similar parasitemia and survival time (WT: 23 ± 2 days, *Tb927.6.4140*-KD: 24 ± 3 days) as WT control mice (Fig. 3c, d, respectively). Next, mice were given doxycycline starting from 3 days post-infection (Fig. 3g). Under these conditions, the effect on first-peak parasitemia and survival (Fig. 3g, h) in *Tb927.6.440*-KD-infected mice was abrogated, indicating that Q586B2 exerts its biological effect within the first 3–4 days of infection.

Collectively, these data illustrate that Q586B2 plays a crucial role in the early stages of *T. b. brucei* infection (before peak parasitemia), allowing a more efficient initiation of infection.

## Q586B2 deficiency promotes early inflammatory responses in vivo and stimulates short stumpy differentiation at peak parasitemia

To mechanistically understand the role of Q586B2 during the early stages of *T. b. brucei* infection, *Tb927.6.4140*-KO, and WT parasites were injected intraperitoneally and 18 h later the CD45+ immune cell composition within the peritoneum was analyzed via flow cytometry (Fig. 4a, and Fig. S7 describing the gating strategy). Within the CD45+ hematopoietic compartment, only the relative presence of macrophages and PMNs was significantly increased in both WT and *Tb927.6.4140*-KO parasite-infected mice as compared to naïve mice (Fig. S8a). Notably, no significant differences in the peritoneal CD45+ immune cell composition were detected between WT and *Tb927.6.4140*-KO parasite-infected mice. While both *Tb927.6.4140*-KO and WT parasite-infected groups showed a drop in the number of peritoneal cells as compared to non-infected animals, significantly less cells were recovered from mice that received *Tb927.6.4140*-KO parasites (Fig. 4b). This excessive drop in peritoneal cell numbers in *Tb927.6.4140*-KO parasite-infected mice was especially reflected in significantly reduced macrophage numbers (Fig. 4b).

A reduction in peritoneal cell numbers, in particular macrophages, has previously been referred to as the macrophage disappearance reaction (MDR)[21], which is indicative of an ongoing local inflammation. Hence, these data suggest that *Tb927.6.4140*-KO parasites may induce a stronger peritoneal inflammation compared to WT parasites. 18 h post-infection, cytokine levels were still very low in the peritoneal fluid, except for MIF, which is known as an inflammatory driver cytokine in AT[22]. Interestingly, *Tb927.6.4140*-KO parasites induced higher MIF levels, corroborating the notion of a stronger peritoneal inflammation at very early stages of infection (Fig S8b). Of note, a similar observation was observed in the serum. To test this assumption further, equal numbers of peritoneal cells from *Tb927.6.4140*-KO and WT parasite-infected mice were cultured, and inflammatory cytokine production was measured. Key inflammatory cytokines, such as TNF, IL-6, and MIF, all of which were documented to play an important role during AT[22–24], were produced at higher levels by peritoneal cells from *Tb927.6.4140*-KO parasite-infected animals compared to WT parasite-infected animals (Fig. 4c). Conversely, the anti-inflammatory cytokine IL-10 was produced less by these peritoneal cells (Fig. 4c).

An alternative or complementary explanation for the reduced peak parasitemia of KO parasite-infected mice could be a higher propensity of these parasites to differentiate into short stumpy forms. To assess this, WT and *Tb927.6.4140*-KO parasites were purified at peak parasitemia and PAD1 expression, a protein only present in stumpy form parasites, was determined via western blot. *Tb927.6.4140*-KO parasites expressed higher PAD1 levels (Fig. S5c), suggesting that *Tb927.6.4140* regulates short stumpy differentiation.

In conclusion, the absence of Q586B2 results in an enhanced inflammatory environment at the site of infection and promotes short stumpy differentiation at peak parasitemia.

## Q586B2 is a secreted protein that is able to induce early IL-10 secretion by myeloid cells in vivo, promoting *T. b. brucei* infection onset

The immunomodulatory effect of Q586B2 may be due to a direct effect on inflammatory cells, provided that this protein is secreted by the parasites. To test this possibility, we developed a Nb39-based sandwich ELISA that was able to detect recombinant Q586B2 in solution (Fig. 5a). Interestingly, Q586B2 was detected in the culture medium (secretome) of in vitro cultured WT parasites, while no signal was obtained for the secretome of *Tb927.6.4140*-KO *T. b. brucei* parasites, indicating that Q586B2 is indeed released by WT parasites (Fig. 5b). Importantly, Q586B2 could also be detected in the serum of *T. b. brucei*-infected mice as early as 6 days post-infection (Fig. 5c), further supporting the conclusion that Q586B2 is released during the early infection phase.

Macrophages are central regulators of inflammation[25], so we next assessed the direct effect of Q586B2 on these cells' ability to secrete pro- and anti-inflammatory cytokines (Fig. 5d). A 48 h in vitro culture of

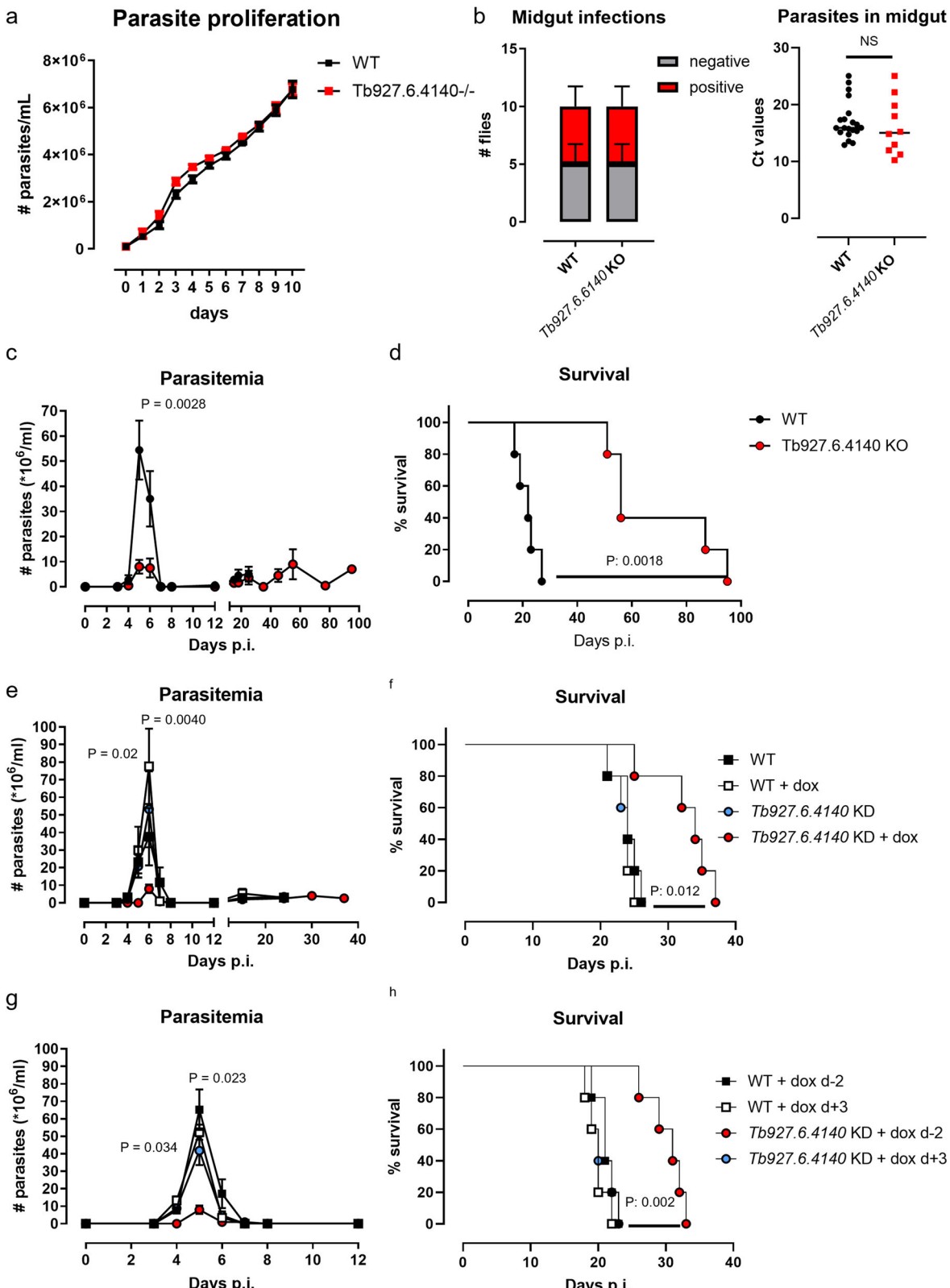

bone-marrow-derived macrophages (BMDMs) with different concentrations of endotoxin-free recombinant Q586B2 protein resulted in a prominent induction of IL-10 secretion by even the lowest Q586B2 concentration, while this was not observed for TNF, IL-6 or MIF. TNF and IL-6 are, however, induced by higher Q586B2 concentrations. Of note, the MIF production by BMDMs stimulated with buffer alone is high, which is in line with findings that BMDMs produce nanogram amounts of MIF upon culturing[26]. Even at intermediate and high Q586B2 concentrations, the level of IL-10 induction was superior to the onset of various inflammatory cytokines (IL-10: 8.4-fold induction at 0.250 μg/mL Q586B2; TNF: 2.9-fold induction; IL-6: 2.6-fold induction; MIF: 1.5-fold induction). Noteworthy, similar results were observed using peritoneal macrophages (Fig. S9). These data suggest that Q586B2 potently induces an anti-inflammatory IL-10-driven response by macrophages.

**Fig. 3 | *Tb927.6.4140*-deficiency does not affect in vitro parasite proliferation and establishment in tsetse flies but attenuates first-peak parasitemia and prolongs survival in the mammalian host. a** In vitro parasite growth of either WT (90:.13 strain) or *Tb927.6.4140*-deficient *T. b. brucei* parasites cultured for 10 days in HMI-9 medium (starting at $5 \times 10^4$ parasites/ml). Results are representative from 3 independent experiments, whereby for each clone and time-point duplicate samples ($n = 2$) were analyzed and data are presented as means ± SEM. **b** Number of *G. m. morsitans* tsetse flies with a midgut infection (procyclic forms, red bars) or with no infection (grey bars). In total, per group, 16 flies fed on blood containing either WT parasites or *Tb927.6.4140*-deficient parasites were evaluated for a midgut infection 10–12 days post-feeding (left panel) and parasite levels were determined via RT-PCR (right panel). NS Not significant. **c, d** Parasitemia and survival of C57BL/6 mice infected with Wild-type (WT: *T. b. brucei* parasite 90:13 pleiomorphic strain, black circles) or *Tb927.6.4140*-deficient (red circles) parasites. Right panels: **e, f** Parasitemia and survival of C57BL/6 mice infected with Wild-type (WT: *T. b.*

*brucei* 90:13 pleiomorphic parasite strain, black boxes), doxycycline-treated WT parasites (white boxes) or *Tb927.6.4140*-knock-down (KD) (blue circles) and doxycycline-treated *Tb927.6.4140*-KD (red circles) parasites. Mice were treated with doxycycline in the drinking water 2 days before initiating the infection.
**g, h** Parasitemia and survival of C57BL/6 mice infected with WT parasites treated with doxycycline 2 days prior to infection (black boxes) or 3 days post-infection (white boxes) as well as *Tb927.6.4140*-KD parasites treated with doxycycline 2 days prior to infection (red circles) or treated after 3 days post-infection (blue circles). Results are representative of 2–3 independent experiments ($n = 5$) and presented as mean (**c, e, g**) or median (**b, d, f, h**) ± SEM. Parasitemia was analyzed using a Multiple paired *t*-test (only significance $p < 0.05$ is shown) and survival curves were analyzed via a Mantel–Cox test (MS median survival). Of note, statistics on parasitemia were performed by comparing doxycycline-treated WT and *Tb927.6.4140*-KD parasites. Source data are provided as a Source Data file.

To assess whether Q586B2 is effectively able to trigger IL-10 production in vivo, we employed IL-10 reporter (VerT-X) mice. 18 h after the intraperitoneal injection of 5 μg endotoxin-free Q586B2, IL-10 production was mainly induced in resident large peritoneal macrophages (LPM) (ΔMFI as compared to PBS control = 255.2) and neutrophils (ΔMFI as compared to PBS control = 240,3) (Fig. 5e and Fig. S7 for the gating strategy of LPM). To test whether the macrophage/neutrophil-derived IL-10 in response to Q586B2 was important to regulate early infection parameters, LysMCre × IL-10$^{fl/fl}$ mice, in which IL-10 is conditionally ablated in macrophages and neutrophils, were infected with WT or *Tb927.6.4140*-KO *T. b. brucei* parasites. Interestingly, no difference in first-peak parasitemia was observed between these two groups of mice (Fig. 5f), in contrast to the reduced peak parasitemia observed in WT mice following infection with *Tb927.6.4140*-KO parasites (Fig. 3c). These results highlight the crucial role of Q586B2-induced myeloid-derived IL-10 to dampen parasite-mediated host inflammation, which in turn promotes first-peak parasitic burden.

Collectively, *T. b. brucei* parasites employ Q586B2 to trigger early IL-10 production by myeloid cells at the site of infection, which promotes a higher first-peak of parasitemia.

### Q586B2 as a potential therapeutic target for trypanosomatids
Since Q586B2 is produced early during the infection and promotes parasite establishment, this protein could be considered as a therapeutic target. Therefore, the panel of anti-Q586B2 Nbs was tested for their ability to block the Q586B2-mediated IL-10 induction in macrophages. To this end, BMDMs were incubated with 1 μg/ml Q586B2 in the presence of 5 μg/ml anti-Q586B2 Nb. Several Nbs were found to significantly inhibit the IL-10-inducing potential of Q586B2, with Nb32, Nb41, and Nb58 being the most potent ones (Fig. 6a). To assess whether these Nbs could affect the infection parameters in vivo, WT mice were injected twice daily with 1 mg Q586B2-blocking or non-blocking Nbs, starting from day 0 until day 4 post-*T. b. brucei* infection. As shown in Fig. 6b, mice treated with a mixture of Q586B2-blocking Nbs (Nb32 and Nb58) exhibited a significantly reduced first-peak parasitemia compared to mice that received a mixture of non-blocking Nbs (Nb39 and Nb53). This Nb-mediated reduction of peak parasitemia is comparable to the phenotype observed in *Tb927.6.4140*-KO and doxycycline-inducible *Tb927.6.4140*-KD *T. b. brucei* parasite-infected mice. Moreover, mice treated with the Q586B2-blocking Nbs survived significantly longer than the control group treated with non-blocking Nbs (Fig. 6c) (median survival time Q586B2-blocking Nb treated: 38 ± 2 days vs non-blocking Nb treated: 32 ± 2 days (*p*-value: 0.01)). These data provide additional evidence that the Q586B2 protein promotes early parasite establishment and propose anti-Q586B2 Nbs as potential therapeutic tools to better control this early infection phase.

### Immunization with Q586B2 attenuates peak parasitemia and prolongs survival in different homologous and heterologous trypanosome models
We next tested whether a Q586B2 immunization could modulate the host response towards the parasite. Recombinant Q586B2 was intraperitoneally (i.p.) injected in the presence of Complete Freund/Incomplete Freund adjuvant, twice at 3 weeks intervals (referred to as Q586B2 immunization), and the subsequent effect on infection parameters was assessed. The control (mock-treated) group included animals receiving only the adjuvant. The Q586B2 immunization triggered a strong humoral immune response against this protein, as demonstrated by the ability of purified IgG antibodies to detect the native Q586B2 protein in parasite lysate via ELISA (Fig. S10a, b). Interestingly, upon challenge (10 days post last dose) with WT *T. b. brucei* parasites, defined as a homologous setting, Q586B2 pre-treated mice showed a significant reduction in first-peak parasitemia and a prolonged survival (mock-treated: 32.6 ± 3.2 days versus Q586B2-treated: 42.4 ± 4.4 days) (Fig. 7a, b).

Considering the high conservation of this protein family within trypanosomatids, the Q586B2 immunization strategy was evaluated for its cross-protective effect against other trypanosomatid species. Interestingly, Q586B2-immunized mice also showed a reduced first-peak parasitemia upon challenge with *T. congolense (*Tc13*)* and *T. evansi*, resulting in a significantly prolonged survival (Tc13: mock-treated: 82.4 ± 8.0 days versus Q586B2-treated: 107.6 ± 18.6 days and *T. evansi*: mock-treated: 16.4 ± 5.3 days versus Q586B2-treated: 25.2 ± 6.2 days) (Fig. 7c–f). Interestingly, the heterologous pre-treatment with Q586B2 also conferred partial protection against one of the most aggressive strains of American trypanosomes, as evidenced by the reduced parasitemia and increased survival of pre-treated mice infected with the aggressive RA strain of *T. cruzi* (Fig. 7g, h).

Together, these results show that an immunization of naïve mice with the *T. b. brucei* protein Q586B2, emulsified in adjuvant, partially protects against a broad range of trypanosomatids.

## Discussion
Parasites acquired elaborate mechanisms to escape the host immune response, allowing parasite establishment and persistence. In this context, many parasites express a MIF homolog that is able to thwart and counteract the mammalian MIF, thereby diminishing MIF-mediated inflammation and elimination by the host immune response. Remarkably, no MIF homolog was found in the genome of African trypanosomes. However, these parasites express an uncharacterized protein, annotated as Q586B2, that presents some MIF-like characteristics, such as a tautomerase activity. Q586B2 was expressed throughout the entire life cycle of *T. brucei*. This is in line with the work from Fergusson et al.[27], who performed a genome-wide comparative proteomic analysis of bloodstream and procyclic *T. brucei* parasites

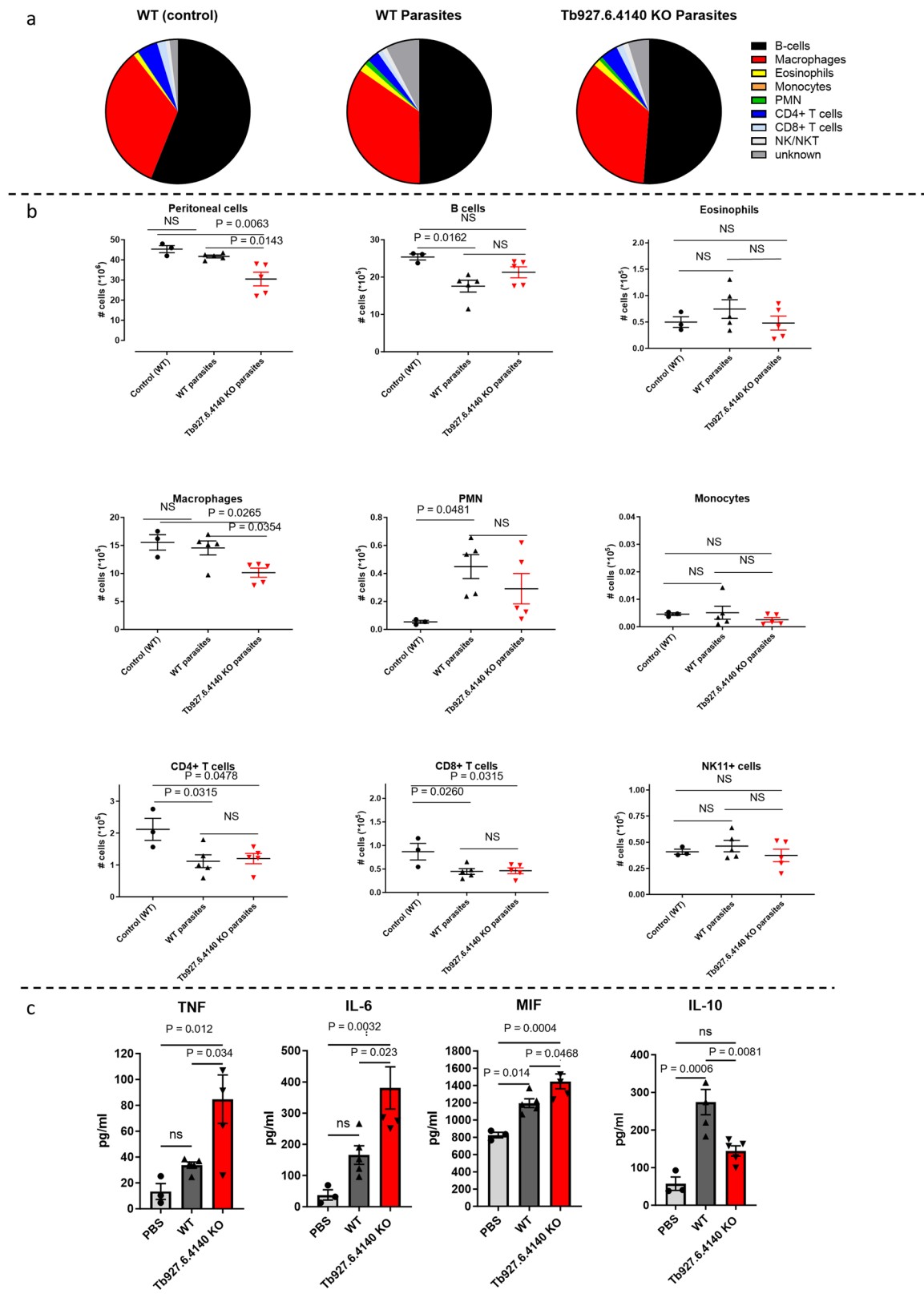

and proposed this protein to be present in both life stages. Fergusson et al. annotated it as "putative paraflagellar rod protein" in the GeneDB (https://www.sanger.ac.uk/tool/genedb/), but no attempts were made to functionally characterize this protein. Our genomic screening and subsequent phylogenetic analysis revealed that this is an evolutionary conserved protein within trypanosomatids, including *T. cruzi* and *Leishmania spp.*, for which there exist 4 paralogues. Since no

homologous genes were found in any other eukaryote taxon, this gene family probably did not originate much earlier than the last common ancestor of *Bodo saltans* and Trypanosomatidae, which lived an estimated 700–460 million years ago[18,28]. This finding has implications for understanding the role of this protein family in the light of parasite evolution and the related adaptation of controlling a host's immune system. First, the lack of homologous proteins outside this clade

**Fig. 4 | *Tb927.6.4140* is essential to attenuate early-stage cellular immune responses within the peritoneum and *Tb927.6.4140*-deficiency induces a stronger anti-inflammatory response within the peritoneum in vitro.** Mice were injected intraperitoneally with PBS (control), 5000 WT (90:13 strain) parasites, or 5000 *Tb927.6.4140*-KO parasites, and 18 h later mice were sacrificed, and the peritoneal cell numbers were determined. Of note, all injections consisted of 200 μl. The cellular composition of the peritoneal cells, i.e., B-cell, eosinophils, macrophages, monocytes, PMN (polymorphonuclear cells), CD4+ and CD8+ T cells, and NK cells, was determined using the gating strategy described in Fig. S5.

**a** Percentages of immune cells withing CD45+ cells in naïve, WT and *Tb927.6.4140*-KO parasite-infected mice. **b** Absolute numbers of total peritoneal cells as well as different immune cell subsets in naïve (control), WT, and *Tb927.6.4140*-KO parasite-infected mice. **c** Production of cytokines (TNF, IL-6, MIF, and IL-10, respectively) by PECs (cultured at $2 \times 10^5$ cells/well) was determined following 48 h incubation. Results are representative of 2 independent experiments ($n = 3$ for the control and $n = 5$ for the infected groups) and presented as mean ± SEM. A one-way ANOVA with Turkey's multiple comparison test was performed. NS not significant.

explains why no Q586B2-related protein was observed in *Plasmodium falciparum*, as this parasite descended from a very different early-diverged eukaryote lineage. *Plasmodium* belongs to the Apicomplexa phylum within the "SAR" supergroup of eukaryotes, while Trypanosomatidae are Euglenozoa within the supergroup Discoba, a clade of 'excavate' eukaryotes supported by phylogenomic data[18,29]. Second, since the *Bodo saltans* sequences are scattered across several of the major gene clades, the duplication events that gave rise to the four paralogous genes in most taxa must have happened before the divergence of *Bodo saltans* and Trypanosomatidae. Consequently, diversification of this gene family most likely happened prior to the origin of an obligatory parasitic lifestyle in Trypanosomatidae. It is now generally accepted that parasitism arose after the divergence of trypanosomatids from free-living flagellates including *Bodo saltans*, but prior to the basal divergence between *Paratrypanosoma confusum* and all other trypanosomatids[18,30–32]. We hypothesize that this protein family originally evolved to serve a currently unknown function unrelated to parasitism. Only later, at least one of these paralogues (represented by Q586B2 in *T. b. brucei*) seems to have acquired a role in the regulation of the host's immune system. This scenario represents a case of evolutionary exaptation, in which one or several proteins evolved a new function alongside a previously existing one. For example, while our study demonstrated a regulatory effect of one member of this protein family on the mammalian immune system, thereby confirming their role in dixenous species during early mammalian host infection, it would be interesting to assess whether members of this protein family are similarly involved in the infection of insect hosts and, thus, may have been recruited earlier to serve an immunomodulatory function in a monoxenous ancestor.

Further analysis demonstrated that Q586B2 was present intracellularly within different vesicles. Indeed, co-localization experiments revealed that Q586B2 partially colocalizes with glycosomes, which is in line with the work by Colasante et al.[33], who identified this protein via nanoLC/MS in the proteome of glycosomes isolated from bloodstream and procyclic *T. b. brucei* parasites that were cultured in vitro. Likewise, Q586B2 was identified as a glycosomal protein within the proteome of procyclic *T. brucei* parasites via epitope-Tag Organelle Enrichment and SILAC quantitative Proteomics[34–36]. The co-localization experiments further revealed that Q586B2 was also present within acidocalcisomes as well as in early endocytic and recycling vesicles. The presence of Q586B2 within acidocalcisomes could be due to its interaction with Q389K9 (*Tb927.10.11300*), an interaction that was identified via yeast-two-hybrid screening (https://www.ebi.ac.uk/intact/search?query=id: Q586B2*#interactor). Q389K9 was indeed shown to be present within acidocalcisomes[37]. The presence of Q586B2 in endocytic and recycling vesicles suggests that Q586B2 can be released, which is in line with our Nb-based ELISA data showing Q586B2 presence in the medium or serum of infected animals.

To acquire more insight into the biological role of Q586B2, we generated *Tb927.6.4140*-deficient parasites. These parasites were fully viable without any effect on cell proliferation in vitro and establishment within the midgut of tsetse flies, suggesting that Q586B2 is not essential for parasite survival. Our data further demonstrate that the early presence of Q586B2, within the first three–four days post-infection, is required to attenuate the host inflammatory response, thereby

allowing efficient parasite establishment and dissemination within its host. Moreover, IL-10 produced by large peritoneal macrophages and/ or PMNs was shown to be instrumental in the Q586B2-mediated parasite establishment. Our in vitro data demonstrate that Q586B2 can induce both pro- as well as anti-inflammatory cytokines, yet it exhibits a more potent effect on IL-10 production, at least in macrophages, via an as yet unknown mechanism. These results are in line with the work of Caljon et al.[38], showing a rapid induction of IL-10 (within the first 18–90 h post-infection) and a decline of the inflammatory response at the site of infection (i.e., the skin) using a natural tsetse fly-mediated infection route. Furthermore, upon skin damage, Tim4+/CD206+ dermal macrophages were reported to produce IL-10, which in turn allows the survival and maintenance of anti-inflammatory Tim4+/IL-10+ macrophages[39]. Hence, the *T. b. brucei*-derived Q586B2 protein appears to be a key mediator in triggering this early IL-10 production by myeloid cells and thereby limiting excessive early inflammatory responses in favor of parasite establishment. This places Q586B2 in a series of *T. brucei*-derived proteins, including the *T. brucei* Kinesin Heavy Chain 1 (TbKHC1) and *T. brucei* Adenylate Cyclase (TbAdC), that can sculpt a more permissive environment for parasite survival[9,19,40]. However, the reduced peak parasitemia in *Tb927.6.4140*-deficient parasite-infected mice could also be partially due to the fact that these parasites are more prone to differentiate into short stumpy forms, as evidenced by their higher PAD1 expression. Hence, Q586B2 could play a dual role during the early stages of infection, whereby on the one hand it could prevent excessive host inflammation essential for parasite establishment, and on the other hand prevent excessive short stumpy differentiation thereby allowing first-peak development. Hence, it is tempting to speculate that Q586B2 could serve as a parasite sensor or regulator for quorum sensing in favor of parasite establishment.

Since Q586B2 is highly conserved amongst trypanosomatids of the Kinetoplastida phylum (including medically important parasite species such as *T. brucei*, *Leishmania*, and *T. cruzi*), and an important virulence factor, we considered it as a potential therapeutic target. In this context, we identified a series of anti-Q586B2 Nbs that could prevent the protein's IL-10-inducing capacity and that could reduce first-peak parasitemia and prolong survival when administered to the infection site shortly after parasite inoculation. Of note, the effect on survival of using blocking Nbs is less pronounced than the effect of knocking-out the gene in *T. brucei*, which could be due to the rapid clearance rate of the Nbs and/or the fact that treatment was halted after 4 days post-infection. However, the pharmacokinetics of the Nb constructs could be improved by reformatting them into a half-life extended format[41] or, alternatively, by injecting the Nbs for a longer period of time (post-peak parasitemia). This passive immunization, though interesting, requires insight in the timing of the bite by a parasite-infected tsetse fly, considering that the Nb-based Q586B2 blockade only works in the earliest stages after infection. Hence, we considered an immunization with Q586B2, which was indeed found to attenuate peak parasitemia and prolong host survival. It is important to realize that the effect on survival in this setting, though significant, may be hampered by the fact that these parasites are able to escape elimination via migration into tissues (dermis, fat tissue, brain) that are less accessible to antibodies[42]. Nevertheless, its

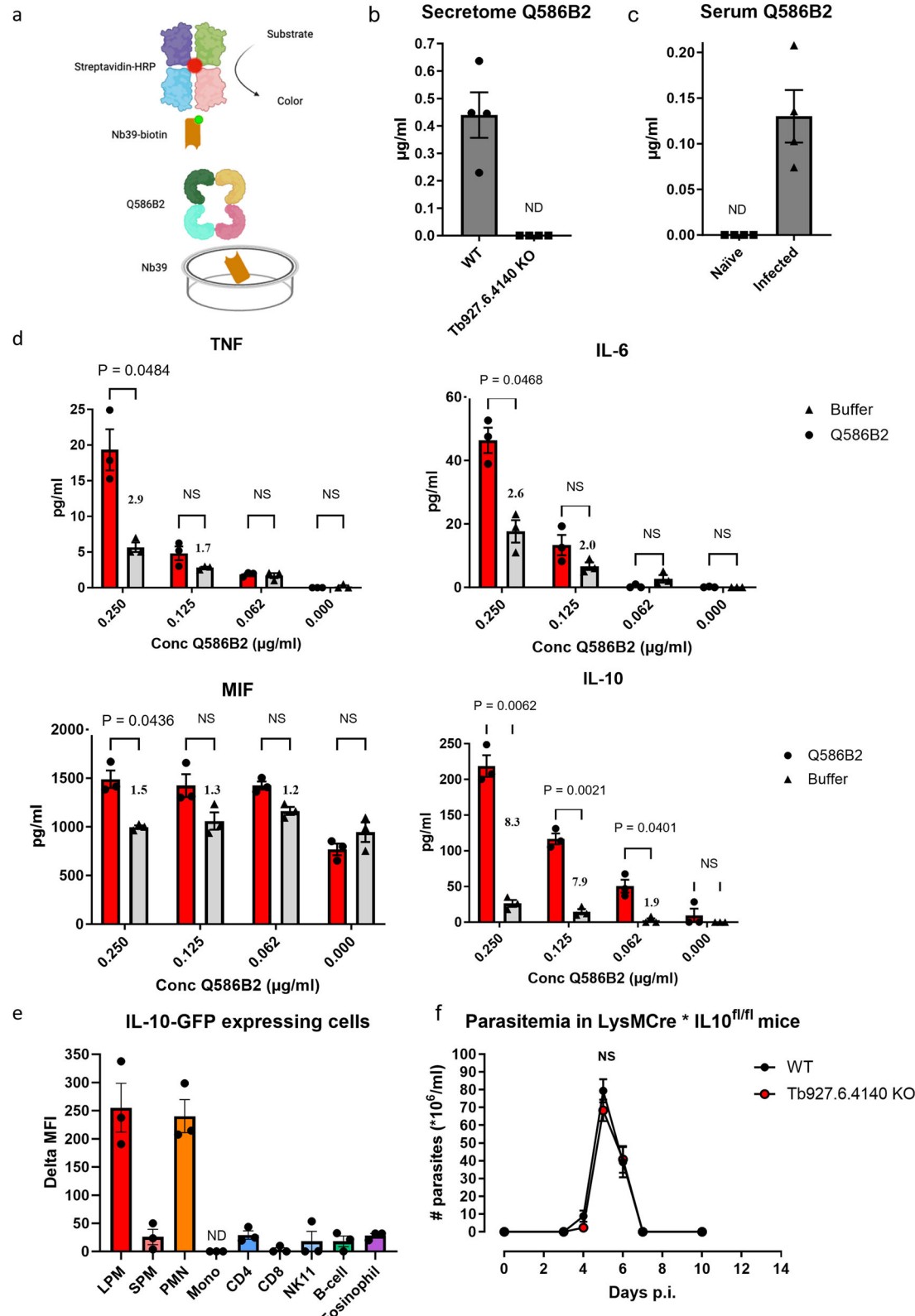

partially protective effect is not only observed in a homologous but also in a heterologous setting, with cross-protection against *T. congolense* and *T. evansi*. These findings resemble our previously reported GPI-based immunization approach, where also cross-protection against heterologous parasites was obtained[43]. Interestingly, this prophylactic Q586B2 immunization also conferred partial cross-protection against an aggressive strain of *T. cruzi*. This is to our knowledge the first time that a conserved molecule was identified in trypanosomes with the ability to induce cross-protection against heterologous parasites as well as against different trypanosomatids. Hence, our findings could pave the way to develop a broad-spectrum intervention strategy for human diseases such as HAT, leishmaniasis, and Chagas disease as well as veterinary and economically relevant diseases such as nagana.

**Fig. 5 | Q586B2 is released by parasites, exerts both pro- and anti-inflammatory activity on myeloid cells (BMDMs) in vitro and at the site of inoculation myeloid cells are the most prominent IL-10 producing cells mediating the attenuated first-peak parasitemia observed in *Tb927.6.4140*-KO parasite-infected mice. a** Schematic of the Nb-based sandwich ELISA whereby Nb39 was used as capturing Nb, biotinylated Nb39 was used as detecting Nbs and streptavidin-HRP was used to develop the ELISA. The illustration in (**a**) was created using Biorender.com by Maida Živalj. **b** Detection of native Q586B2 via a Nb-based sandwich ELISA in secretome of WT (90:13 strain) and *Tb927.6.4140*-KO parasites cultured in HMI-9 for 48 h in vitro. **c** Detection of native Q586B2 via a Nb-based sandwich ELISA in serum from naïve and *T. b. brucei*-infected C57BL/6 mice collected at day 6 p.i. For **b** and **c**, results are representative for 2 independent experiments (*n* = 4) and presented as mean ± SEM. Unpaired *t*-test (Two-tailed) was performed using Welch's correction. **d** Production of TNF, IL-6, MIF, and IL-10 by BMDMs generated from C57BL/6 mice (2 × 10⁵ cells/well) following 48 h incubation with a ½ serial dilution of LPS-free Q586B2 (red bars) starting from 0.25 μg/ml or buffer alone (gray bars). Negative controls: cells incubated with buffer alone or

without any protein. Data are shown as means ± SEM and representative of 2 independent experiments (*n* = 3). A Multiple paired Student's *t*-tests was used for comparison between experimental groups, with ns indicating not significant for *p* ≥ 0.05 and *p* < 0.05. **e** To assess which cells express IL-10 in vivo, IL-10-GFP (VeRT-X) mice were injected intraperitoneally with PBS (control) or 5 μg LPS-free Q586B2 and 18 h later sacrificed. Subsequently, the cellular source of IL-10 was analyzed via flow cytometry using the gating strategy described in Fig. S5. The expression of IL-10 was presented as delta median fluorescence intensity (delta MFI), by subtracting the MFI of PBS-injected mice from Q586B2-injected mice, for each of the different immune cells that were identified. Results are representative of 2 independent experiments (*n* = 3) and presented as mean ± SEM. ND Not detected. **f** Parasitemia of LysMCre × IL-10^fl/fl^ C57BL/6 mice infected with Wild-type (WT: *T. b. brucei* parasite 90:13 pleiomorphic strain, black circles) or *Tb927.6.4140*-deficient (red circles) parasites. Results are representative of 2 independent experiments (*n* = 5) and presented as mean ± SEM. NS not significant, ND Not detectable. Parasitemia was analyzed using a Mann–Whitney *U*-test (only significance *p* < 0.05 is shown). Source data are provided as a Source Data file.

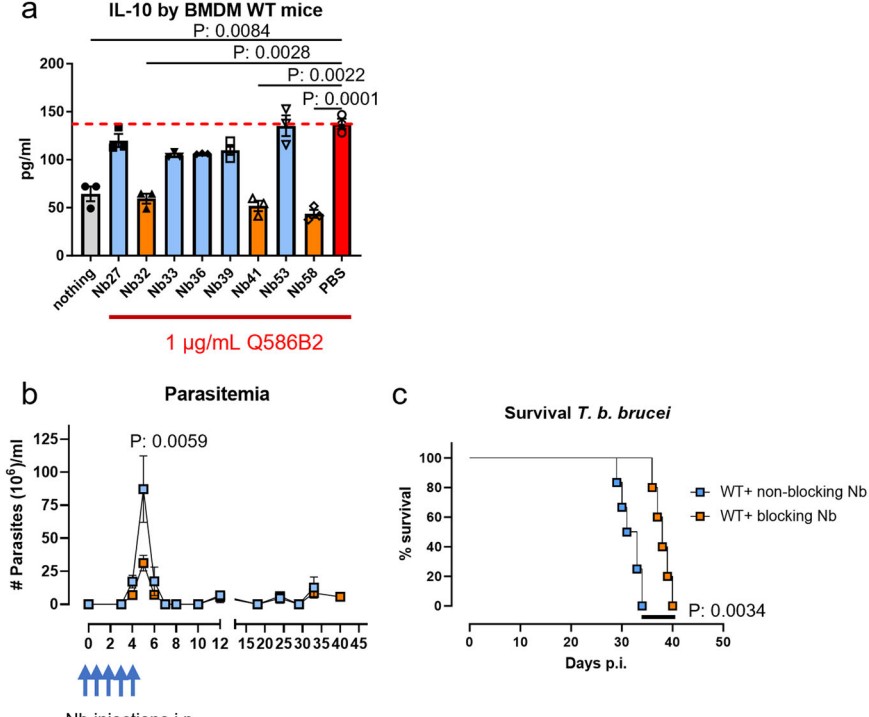

**Fig. 6 | Anti-Q586B2 Nbs exert a Q586B2-blocking potential on myeloid cells (BMDMs) in vitro as well as a therapeutic potential. a** BMDMs generated from C57BL/6 mice (2 × 10⁵ cells/well) were incubated with 1 μg/ml of LPS-free Q586B2 (white bars) or buffer alone (black bars in presence or absence of anti-Q586B2 Nbs (5 μg/ml)). After 48 h incubation the culture medium was tested for IL-10 production. Negative controls: cells incubated with buffer alone or without any protein. Dashed red line represent levels of cytokines produced by BMDMs stimulated with Q586B2 alone and all groups were compared to this signal. Data are shown as means ± SEM and representative of 2 independent experiments (*n* = 3). Analysis was performed using a one-way ANOVA with Brown–Forsythe and Welch's test (only

significance *p* < 0.05 is shown). **b, c** Parasitemia development and survival of C57BL/6 mice infected with Wild-type *T. b. brucei* AnTat1.1E parasites and treated with non-blocking anti-Q586B2 Nbs (blue box) or blocking anti-Q586B2 Nbs (orange box) starting from days 0 post-infection and this for 4 consecutive days (indicated by blue arrows). Results are presented as mean (parasitemia) or median (survival) of 5 individual mice ± SEM. Parasitemia was analyzed using a Mann–Whitney *U*-test (only significance *p* < 0.05 is shown) and survival curves were analyzed via a Mantel–Cox test (MS median survival). Source data are provided as a Source Data file.

## Methods

### Ethics statement
All experiments complied with the ECPVA guidelines (CETS n° 123) and were approved by the VUB Ethical Committee (Permit Number: 17-220-02). The protocol of animal procedures followed in this study was approved by the Ethics Committee of Animal Experiments of the Universidad Nacional de San Martín (CICUAE N° 14/2022). Mice were monitored daily. Humane endpoints were used during the study, based on weight loss, animals with >25% weight loss were sacrificed using carbon dioxide treatment.

### Mice, parasites, and infections
Eight-week-old female C57BL/6 mice were purchased from Janvier, France. Vert-X (B6(Cg)-*Il10^tm1.1Karp*/J) were purchased from Jackson Laboratory, USA. *LysMCre × IL-10^fl/fl* (i.e., LysM-IL-10) mice were generated in-house by crossing the *LysMCre* (B6.129P2-*Lyz2^tm1(cre)Ifo*/J, JAX stock #004781) mice with *IL-10^fl/fl* mice (a kind gift of W. Muller, University of Manchester, Manchester, United Kingdom), as described before[44].

Bloodstream trypanosome parasites were stored at −80 °C as blood aliquots containing 50% Alsever's solution (Sigma–Aldrich) and

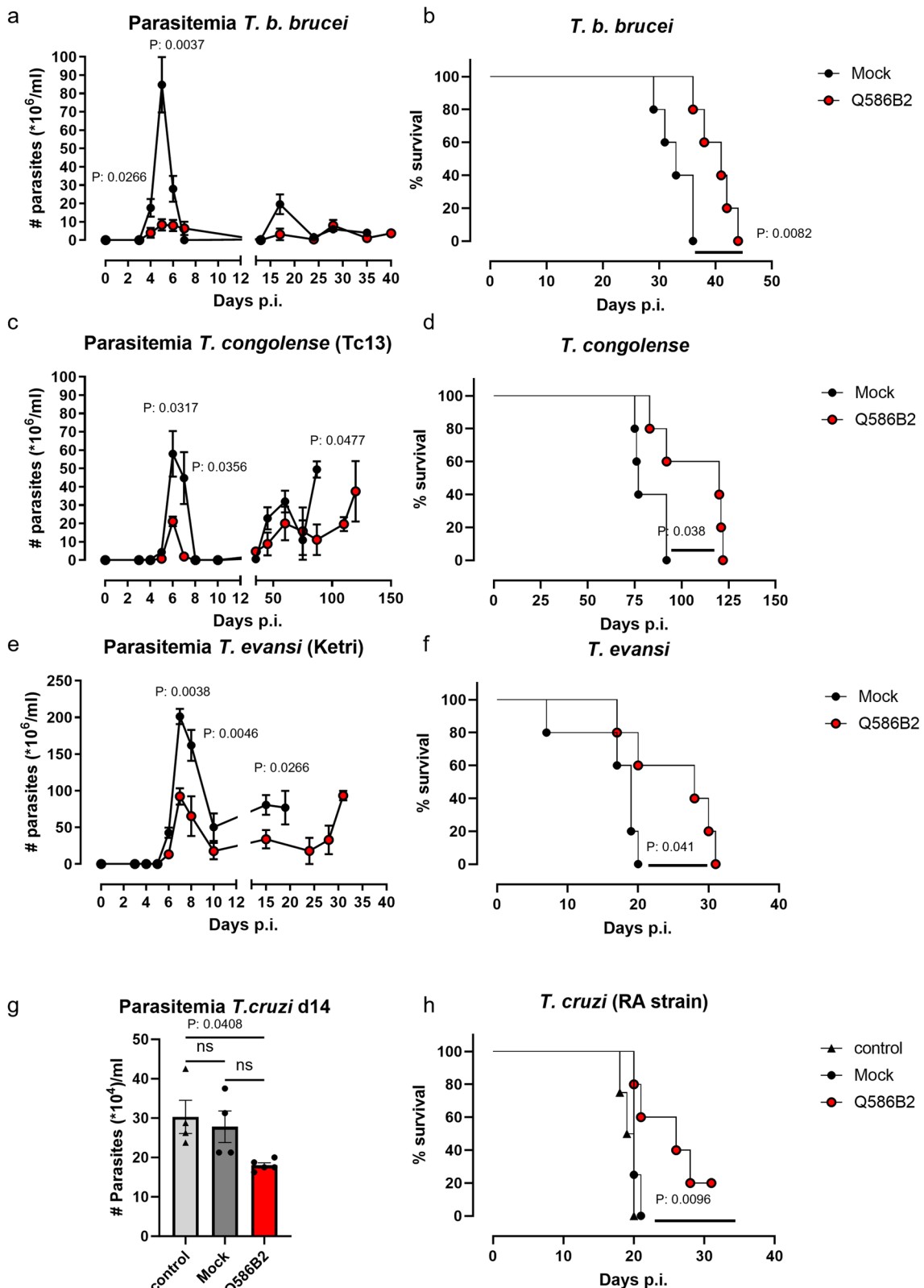

**Fig. 7 | Q586B2 can be considered a broad-spectrum target for trypanosomes.**
**a**, **b** Parasitemia development and survival of mock-treated (black circles) and Q586B2-treated (white circles) C57BL/6 mice infected with Wild-type *T. b. brucei* (AnTat1.1E) parasites. **c**, **d** Parasitemia development and survival of mock-treated (black circles) and Q586B2-treated (red circles) C57BL/6 mice infected with *T. congolense* parasites. **e**, **f** Parasitemia development and survival of mock-treated (black circles) and Q586B2-treated (red circles) C57BL/6 mice infected with *T.*

*evansi* parasites. **g**, **h** Parasitemia development and survival of mock-treated (black circles) and Q586B2-treated (red circles) C57BL/6 mice infected with *T. cruzi* parasites (RA strain). Results are presented as mean (parasitemia) or median (survival) of individual mice ($n = 5$) ± SEM. Parasitemia was analyzed using a Mann−Whitney $U$-test (only significance $p < 0.05$ is shown) and survival curves were analyzed via a Mantel−Cox test (MS median survival). Source data are provided as a Source Data file.

10% glycerol (final V/V). Parasites used: clonal pleomorphic *T. brucei* AnTat1.1E parasites were a kind gift from N. Van Meirvenne (Institute for Tropical Medicine, Belgium), clonal *T. congolense* parasites (Tc13) were kindly provided by Dr. Henry Tabel (University of Saskatchewan, Saskatoon), clonal *T. evansi* (KETRI2479, Kenia), *T. brucei gambiense* (Feo, Prof. Vincendeau, France), *T. brucei rhodesiense* (ETat 1.2S, Uganda) and *T. vivax* (ILARD700/U700, Institute of Tropical Medicine, Belgium). In experiments where *Tb927.6.4140*-KO and *Tb927.6.4140*-KD parasites were used, the pleomorphic *T. b. brucei* cell line 90:13 (Engstler et al.[45]) was used as control since this is the parental clone. Female mice (7–8 weeks old) were infected with $5 \times 10^3$ trypanosomes via intraperitoneal (i.p.) injection in a volume of 200 μL PBS. To knockdown the *Tb927.6.4140* gene in vivo, mice received doxycycline (1 mg/ml in drinking water + 5 g/l fructose, bottle wrapped in aluminum foil), two days before infection or at day 3 post-infection. The solution was replaced with fresh medium each two days. Parasite numbers in blood were determined via hemocytometer by tail-cut (2.5 μl blood in 500 μl RPMI).

Q586B2-vaccination consisted of i.p. injection of 5 μg of LPS-free Q586B2 dissolved in 100 μL PBS and 100 μL Complete Freund Adjuvant (CFA, ThermoFisher Scientific), followed 3 weeks later again with the same material in 100 μl Incomplete Freund Adjuvant (IFA, ThermoFisher Scientific). As control only PBS in either CFA or IFA was used. Ten days post last injection 2.5 μl blood was collected via tail-cut and diluted in 500 μl RPMI for ELISA measurement to determine the antibody titer.

For *T. cruzi* infections, BALB/cJ male mice (60-day-old) were immunized as described above. Immunized, mock immunized, and naïve control animals were challenged i.p. with 50 parasites from the RA strain[46], routinely maintained by serial passages in CF1 mice. Parasitemia was tested by sampling blood from the tail tip and counting in a Neubauer chamber. Mice were monitored during 60 days after infection.

For the in vivo Nb treatment experiment mice, wild-type *T. b. brucei* parasite-infected mice were injected i.p. twice daily with non-blocking anti-Q586B2 Nbs (Nb39 and Nb53) or blocking anti-Q586B2 Nbs (Nb32 and Nb58) (1 mg/mouse/injection) starting from days 0 post-infection and this for 4 consecutive days. Parasitemia at different time points were examined as outlined above.

## Trypanosoma cell lines and constructs

The pleomorphic *T. b. brucei* cell line 90:13 (Engstler et al.[45]) was cultivated in culture medium (HMI-9 supplemented with 15% FCS) at 37 °C and 5% $CO_2$. Transfections were performed with $2 \times 10^7$ trypanosomes as described by Burkard et al.[47], and drug selection was applied 6 h post-transfection on parasites diluted to $2 \times 10^4$ cells/ml (phleomycin 3 μg/ml; puromycin 0.5 μg/ml). Resistant clones were analyzed by PCR on gDNA (Zymo Research).

For the RNAi constructs, the full *Tb927.6.4140* coding sequence was amplified by PCR and cloned in p2T7-TiTA plasmid as described in Wickstead et al.[48] For the KO constructs, the 500 bp genomic sequences upstream and downstream of the *Tb927.6.4140* coding sequence was amplified by PCR from genomic DNA and subcloned in pUC KO Phleomycin and Puromycin plasmids as described by Uzureau et al.[49] Primers used for the KO constructs: -5′ UTR Forward: TTGGGCTCGAGTTCACAAGTG, Reverse: CGGTCAGATCCATGGTCTAGA TTTTTTTTCTTTTTTCTTTCCTCCTTGTTTTATTGTC (Puromycin); GTCAACTTGGCC ATGGTCTAGATTTTTTTTCTTTTTTCTTTCCTCCTT GTTTTATTGTC (Phleomycin); -3′UTR Forward: CGAGCTCGGTACC CGGGGATCCTAGTGGCCAGTGTATTTGGG CCG; Reverse: TGTGAACT CGAGCCCAAAATGCTCATATACGC. Primers used for the RNAi/P2T7 constructs: Forward: ACGTACGGATCCATGCCAAACCTCGTTGTC, Reverse: GTACGTCTCGAGGCCACTAAGAGCGAGTG. Plasmids were linearized before transfection. Double-stranded RNA production was induced by addition of 1 μg/ml doxycycline. Primers used for Fig. S4b;

*Tb927.6.4140* Forward: 5′- ATGCCAAACCTCGTTG TC-3′; Reverse 5′-GAGCGAGTGAAATAAAACT-3′; *tubulin* Forward: 5′-ATGTGTACTT TGATGAGG-3′; Reverse: 5′-TGACGCCGGACACAACAG-3′. cDNA PCR conditions: 95 °C for 3 min, (94 °C for 30 s, 57 °C for 30 s, 72 °C for 45 s) in total 28 cycles, 72 °C for 10 min. For the in situ V5-tagging transfections were performed whereby cells are amplified to $2 \times 10^7$ per transfection. Trypanosomes are centrifuged at RT, $1569 \times g$ for 10 min and then resuspended in Amaxa transfection buffer (Human T Cell Nucleofector Kit, Lonza). The resuspension is incubated with 20 μl resuspension of PCR long primer (for in situ tagging). Primers used see Table S2. Electroporation of trypanosomes is performed using the Nucleofector™ 2b Device (Amaxa) and the Z-001 program. The electroporated cells are put in the presence of selection drug (5 μg/ml blasticidin) after 6 h of incubation in culture medium.

## Parasite purification

Eight-week-old C57Bl/6 mice were injected intraperitoneally with 5000 *Trypanosoma* parasites (*T. brucei brucei* (AnTat1.1E), *T. brucei gambiense* (Feo), *T. brucei rhodesiense* (ETat 1.2S, Institute of Tropical Medicine, Belgium), *T. congolense* (Tc13), *T. evansi* (KETRI2479), and *T. vivax* (U700)) intraperitoneally. Five to six days after injection (i.e., peak parasitemia) mice were euthanized with $CO_2$ after which the heparinized blood was taken via cardiac puncture. A total volume of 450 μl blood was taken with a syringe equilibrated with 50 μl heparin (Sigma–Aldrich, 1/10 dilution in PBS (1000 units/ml)). The parasites were purified from the blood via a DEAE-52 anion-exchange column (Whatman) equilibrated against phosphate saline glucose (PSG) buffer as described before[50]. The procyclic and metacyclic AnTar1 parasites, *T. cruzi*, *Leishmania infantum*, and *Leishmania major*, and *Plasmodium falciparum* were provided by Prof. Guy Caljon (University of Antwerp, Belgium).

Parasites were brought at a concentration of $10^7$/ml in PSG buffer, centrifuged at $805 \times g$ (5180R Eppendorf Centrifuge) at room temperature, resuspended in 200 μl BD Cytofix/Cytoperm™ (Fisher Scientific) and kept at 4 °C for 20 min. Finally, the cells were washed with BD Perm/Wash buffer (Fisher Scientific) (1 ml/tube), centrifuged at $805 \times g$, resuspended in 1 ml Perm/Wash buffer, and stored in the cold room at 4 °C for further analysis (Immunofluorescence microscopy or Flow cytometry).

## In vitro parasite cell cultures

*T. brucei brucei* (90:13) and *Tb927.6.4140*-KO parasites were grown in vitro in a 6-well plate with 3 ml HMI-9 medium and 20% heat-inactivated Fetal Bovine Serum (iFBS). Parasites were seeded at a concentration of $4 \times 10^5$ cells/mL in 2 mL of HMI-9 with 10% iFBS. Cell growth was monitored by microscopic counting using a Neubauer-improved hemocytometer for a period of 9 days. Cells were diluted 1:5 daily. Alternatively, the *T. brucei brucei* (90:13) and *Tb927.6.4140*-KD parasites were grown at 37 °C and 5% $CO_2$ in presence of geneticin + hygromycin (1 μg/ml each) for the 90:13 parasites and with geneticin +hygromycin+ phleomycin (1 μg/ml each) and in presence of doxycycline (Duchefa, 1 μg/ml) for the *Tb927.6.4140*-KD parasites and medium refreshed every 2–3 days. Culture media were kept at −20 °C for further analysis. Parasite numbers were determined microscopically.

## Tautomerase assay

MIF tautomerase assay was performed as previously described[51]. Briefly, the enzymatic reaction was initiated at 25 °C by adding 20 μl of dopachrome methyl ester substrate (2 mM L-3,4-dihydroxyphenylalanine methyl ester and 4 mM sodium periodate) in a 96-well plate that contained 200 μl of either (i) rmMIF (83.3 nM, R&D systems), (ii) rmMIF preincubated for 1 h at 37 °C with 10 μM ISO-1 (Merck, Darmstadt, Germany), (iii) Q586B2 (83,3 nM), (iv) Q586B2 preincubated for 1 h at 37 °C with 10 μM ISO-1, or (v) BSA. All dilutions were made in tautomerase assay buffer (50 mM potassium phosphate, 1 mM

EDTA, pH 6.0). Activity was determined by the semicontinuous reduction in signal that was measured at $OD_{475nm}$ for 5 min. The percentage tautomerase activity was compared to that of rmMIF alone (set at 100% activity). ELISA plates were read by using an $EL_X808$ Absorbance Microplate Reader (BioTek Instruments, Winooski, VT, USA) and Gen5 1.08 software (BioTek Instruments).

### Immunization, library construction, and screening

Immunization, library construction, and screening were performed as previously described[52]. In brief, Nb libraries were generated by using peripheral blood lymphocytes isolated from a llama that was immunized 6 consecutive times with 100 µg recombinant Q586B2 (purification performed by the VIB Protein Core (VIB, Belgium)) in presence of the same volume of Gerbu adjuvant. The Nb phage library was constructed after total RNA extraction and cDNA subcloning into the phagemid vector pMECS. As soon as enrichment was observed on recombinant Q586B2 (10 µg/ml), individual colonies were selected and transformed into chemo-competent *E. coli* WK6 cells to allow large-scale purification. Eight distinct Nbs belonging to 8 different families based on the difference in their CDR3 regions could be retrieved and the Nb sequences are shown in Fig. S4a(i). Expression and purification of Nbs (encoding a C-terminal HA- and His₆-tag) was similar as described earlier[53]. Briefly, following periplasmic extraction, Nbs were purified using immobilized metal affinity chromatography on a His-Trap column (GE Healthcare, Waukesha, WI, USA) and eluted with 0.5 M imidazole (Sigma–Aldrich) in PBS (pH 7.5). The eluate was subsequently purified on a Superdex 75 (10/300) gel filtration column (GE Healthcare) equilibrated with PBS using an Akta Explorer 10S (GE Healthcare using Unicorn 5.1 software). Next, the purity of the Nbs was evaluated via 12% Bis/Tris SDS-PAGE analysis, and the protein concentration determined via Nanodrop (Nanodrop ND-1000, Isogen, Hackensack, NJ, USA). All Nbs were treated with ProSep-Remtox beads (ImmunoSource, Schilde, Belgium) and confirmed to be LPS-free by the Limulus Amebocyte Lysate Kinetic-QCL Kit (Cambrex, East Rutherford, NJ, USA) in accordance with the manufacturer's instructions. Nbs were either used immediately or stored at −20 °C.

### Cell isolation and culturing

Bone-marrow cells were isolated from tibia and femur of 7- to 8-week-old female WT mice (Janvier, Le Genest-Saint-Isle, France) and following ACK lysis buffer (0.15 M NH₄Cl, 1.0 mM KHCO₃, 0.1 mM Na₂-EDTA) treatment, counted, brought at a concentration of $10^6$ ml, and cultured for 13 days in RPMI-1640 medium supplemented with 300 µg/ml L-glutamine (Sigma–Aldrich), 100 mg/ml streptomycin (ThermoFisher Scientific), 100 units/ml penicillin, 20% (v/v) heat-inactivated FBS (Capricorn Scientific) and 30% L929-conditioned medium. Subsequently, bone-marrow-derived macrophages (BMDMs) were harvested and resuspended in ME-medium ((RPMI-1640 medium, 10% FBS, 1% sodium pyruvate (Gibco), 1% non-essential amino acids (Gibco), 1% glutamate, 1% penicillin-streptomycin, 1% beta-mercaptoethanol)). Ex vivo peritoneal exudates cells (PECs) were obtained from $CO_2$-euthanized 8-week-old female C57BL/6 mice (Janvier, Le Genest-Saint-Isle, France). Cells were harvested (*i.e.*, peritoneal lavage) by using 10 ml ice-cold PBS and subsequently centrifuged (Eppendorf Centrifuge 5810R) at $394 \times g$ for 7 min at 4 °C to be resuspended in ME-medium. Of note, the peritoneal lavage of naïve mice as well as trypanosome-infected mice was kept to measure cytokine levels (see further), following concentration using Vivaspin concentrators (MWCO-3.5 kDa). Cells were diluted at $2 \times 10^6$ cells/ml in ME-medium, after which 100 µl of cell suspension/well (BMDMs) or 500 µl of cell suspension/well (PECs) were cultured for 48 h (37 °C, 5% $CO_2$) in 96- or 48-well plates (Nunc), respectively. For the BMDMs, cells were stimulated with either (i) a ½ serial dilution of LPS-free Q586B2 starting from 0.25 µg/ml or (ii) LPS-free Q586B2 (1 µg/ml) in presence or absence of anti-Q586B2 Nbs

(5 µg/ml). As negative control the same volume of buffer as for Q586B2 was used. Finally, the supernatant was collected and tested in ELISA.

### Flow cytometry

To analyze the binding of Nbs to fixed and permeabilized parasites, 100 µl of a stock solution of $10^7$ parasites/ml were added to the FACS tubes (Falcon). To each tube, 5 µg of HA-tagged Nb was added after which they were incubated on ice for 30 min Of note, an anti-VSG Nb (Nb33) was used as positive control for Nb binding to parasites[50]. The tubes were washed with 200 µl Perm/Wash and centrifuged for 6 min at $805 \times g$ and 4 °C. To each tube, 1 µg of an Alexa Fluor 488 labeled anti-HA IgG (BioLegend) (1/1000 dilution in Perm/Wash buffer) was added after which they were incubated on ice for 20 min. As negative control parasites in presence of the Alexa Fluor 488 labeled anti-HA IgG alone was used. Finally, 200 µl Perm/Wash was added to the tubes and centrifuged for 7 min at $805 \times g$ and 4 °C. The supernatant was removed and the pellet was vortexed after which 150 µl Perm/Wash buffer was added and the samples measured on a FACSCanto™ II flow cytometer (BD). Data were analyzed using FlowJo software 10.

To analyze the peritoneal cellular composition, $10^6$ cells (in 100 µl) were incubated (15 min, 4 °C) with Fc-gamma blocking antibody (2.4G2, kind gift from Lois Boon (JJP Biologicals)), and subsequently stained with fluorescent conjugated antibodies for 30 min at 4 °C. Fluorescent antibodies used: APC/Cyanine7 anti-mouse CD45 (BioLegend, clone 30-F11), PE/Cyanine7 anti-mouse/human CD11b (BioLegend, clone M1/70), Brilliant Violet 510™ anti-mouse/human CD11b (BioLegend, clone M1/70), APC anti-mouse Ly-6C (BioLegend, clone AL-21), PerCP/Cyanine5.5 anti-mouse Ly-6G (BioLegend, clone 1A8), PE-Cy™7 Rat Anti-Mouse Ly-6G (BD Bioscience, clone 1A8, 560601), PE anti-mouse F4/80 (BioLegend, clone BM8), FITC anti-mouse F4/80 (BioRad, clone Cl:A3-1), Brilliant Violet 421™ anti-mouse I-A/I-E (BioLegend, clone M5/114.15.2), Brilliant Violet 510™ anti-mouse CD8a (BioLegend, clone 53-6.7), PE-Cyanine7 anti-mouse CD90.1 (ThermoFisher Scientific, clone Thy-1.1 (HIS51)), Brilliant Violet 510™ anti-mouse CD19 (BioLegend, clone 6D5), FITC anti-mouse CD4 (BioLegend, clone RM4-5), PE rat anti-mouse NK1.1 (ThermoFisher Scientific, clone PK136), PE rat anti-mouse SiglecF (BD Bioscience, clone E50-2440). All antibodies used are from BioLegend unless stated different as used at a dilution specified by the supplier. Following washing with FACS buffer the cell suspensions were analyzed on a FACSCanto II flow cytometer (BD Biosciences) and data was processed using FlowJo software 10 (Tree Star Inc., Ashland, OR). Briefly, the results were analyzed after exclusion of aggregated and death cells (7AAD⁺, BD Pharmingen) and selection of CD45⁺ cells (leukocytes). The total number of cells in each population was determined by multiplying the percentages of subsets within a series of marker negative or positive gates by the total cell number determined via microscopy counting with trypan blue.

### Fluorescence microscopic analysis

The labeled parasites used in flow cytometry were subsequently analyzed by fluorescence microscopy. Cells were in addition stained with DAPI (Sigma–Aldrich) by the use of the Fluoro Gel II mounting medium (Electron Microscopy Sciences) as described before[54]. For the co-localization experiments, *T. b. brucei* (90:13 strain) trypanosomes from a 24 h culture are centrifuged at room temperature, $1569 \times g$ for 5 min (Eppendorf centrifuge 5810R). Cells are fixed with 3.5% paraformaldehyde-TBS and plated on poly-L-lysine (1 mg/ml) overnight at 4 °C. The cells are permeabilized in 0.1% TBS Triton X-100 for 3 min at room temperature (RT). Next, the sample is saturated in 3% BSA for 30 min at RT. The coverslips are then incubated in the presence of the primary antibody (1:100 α-V5 goat, 1:1000 α-p67 mouse, 1:500 α-RAB2B rabbit, 1:500 α-RAB5B rabbit, 1:500 α-RAB11 rabbit, 1:500 VIT rabbit, 1:500 α-GK rabbit and 1:1000 α-BiP rabbit from James Bangs University of Buffalo and 1:50 Nb39) in 1% BSA-TBS 2 h at 37 °C. The coverslips are washed 4 times for 5 min in TBST (0,05%) at RT and

then incubated in the presence of fluorochrome-conjugated secondary antibodies. For detection, of V5, p67, and BiP, a Donkey anti-Goat IgG (H+L) IgG Cross-Adsorbed Secondary Antibody PE (ThermoFisher Scientific), Goat anti-Mouse IgG1 Cross-Adsorbed Secondary Antibody PE (ThermoFisher Scientific) or Goat anti-Rabbit IgG (H+L) Cross-Adsorbed Secondary Antibody PE (ThermoFisher Scientific) was used respectively (at 1/1000 dilution in 1% TBST), while for the Nb an Alexa-488 labeled anti-HA IgG (diluted 1/1000 in 1% TBST) was used. After 1 h incubation at 37 °C, samples are washed 4 times for 5 min in TBST. The coverslips are mounted with 4 µl of ProLong Gold Antifade with DAPI mounting medium (Invitrogen). The slides were observed using a Zeiss Axio Imager M2 wide field.

## PAD1 expression experiments

*T. brucei brucei* (90:13) and *Tb927.6.4140*-KO parasites were isolated at peak parasitemia and purified using DEAE-resin (as described in Stijlemans et al.)[50], after which the parasites were loaded on an SDS-page under reducing conditions and transferred to PVDF membrane. Next, PAD1 and EF1α protein expressions were determined as described in Mabille et al.[55] and data quantified (i.e., densitometry) using a Vilber Fusion Imager (Viber) and Fusion FX6 Edge 18.05 software. Primary antibodies for PAD1 and EF1α (kindly gifted by Prof. Keith Matthews, University of Oxford) were used at 1:1000 and 1:7000, respectively. For detection, an HRP-coupled goat anti-Rabbit IgG (H+L) (1/1000, ThermoFisher Scientific) or goat anti-Mouse IgG (H+L) (1/1000, NovusBio) were used.

## Quantification of cytokines and antibodies

All cytokines except MIF were quantified using the V-PLEX Custom Mouse Cytokine kit (catalog number K152A0H) from Meso Scale Discovery (MSD, Rockville, MD, USA) according to the manufacturer's protocol. Alternatively, the culture medium concentrations of MIF, TNF, and IL-10 (R&D Systems) as well as IL-6 (Pharmingen) were determined by ELISA as recommended by the suppliers. The cytokine levels in peritoneal lavage were determined following concentration (10 times) using Vivaspin concentrators (MWCO-3.5 kDa). Anti-Q586B2 IgGs present in the serum of Q586B2-vaccinated mice were determined via ELISA. Briefly, Q586B2 was coated overnight at 1 µg/ml PBS in 96-well MaxiSorp plates (NUNC). Plates were washed (0.1% Tween 20 in PBS) and blocked (1% BSA in PBS) for 1 h. Next, plates were washed and the sera from individual mice were serially diluted starting from 1/100 in blocking buffer. The ELISA was subsequently performed as described by the suppliers (SBA Clonotyping system-HRP kit (SouthernBiotech, USA)). As negative controls, serum from mock-vaccinated mice (a pool of 5 naive mice) was used as well as plates without Q586B2. The $OD_{450nm}$ recorded on Q586B2-free plates was subtracted from the $OD_{450nm}$ recorded on the Q586B2-coated plates. To further assess if the IgG present in the serum could recognize specifically the native antigen in parasite lysate, IgGs from pooled mice were purified using protein-G chromatography (ThermoFisher Scientific) as described by the suppliers, the concentration determined spectrophotometrically and then stored at −80 °C till further use. Subsequently, the IgGs were tested in ELISA on coated lysate from *T. b. brucei* (10 µg/well) prepared as described in Obishakin et al.[56] To this end, following blocking the plate with blocking buffer (1% BSA in PBS), the purified IgGs were ½ serially diluted in blocking buffer starting at 1 µg/ml. The rest of the procedure is the same as for testing the serum in ELISA (see above).

## Nb-based Q586B2 ELISA

Nb39 was coated overnight at 5 µg/ml PBS in 96-well MaxiSorp plates (NUNC). Plates were washed (0.1% Tween 20 in PBS) and blocked (1% BSA in PBS) for 1 h. Next, plates were washed and culture medium from in vitro cultured parasites or serum from naïve and infected (day 6 p.i.) mice (diluted 1/2 in blocking buffer) were added. A standard curve was made with recombinant Q586B2 starting at 10 µg/ml (serially diluted ½ in blocking buffer). After 2 h incubation at 37 °C, the plates were washed 5 times with 0.1% Tween 20 in PBS and a biotinylated Nb39 (5 µg/ml in PBS) was used as detecting agent. Two hours later, plates were again washed and streptavidin-HRP was added for one hour. After a final washing, the $OD_{450nm}$ was recorded and following subtraction of the background signal (wells without Q586B2 added or HMI-9 culture medium alone) the concentration of Q586B2 was determined.

## Aspartate aminotransaminase (AST) and alanine aminotransaminase (ALT) measurement

Serum AST and ALT levels were determined as described by the suppliers (Sigma–Aldrich).

## Anemia determination

Red blood cell (RBC) numbers in blood were determined via hemocytometer by tail-cut (2.5 µl blood in 500 µl RPMI) after which anemia was expressed as percentage of RBCs remaining in infected mice compared to that of non-infected mice.

## Statistical analysis

The GraphPad Prism 7 software was used for statistical analyses, Student *t*-test for paired analyses or one-way ANOVA (with Turkey's multiple comparisons test) or two-way ANOVA; and Log-rank (Mantel–Cox test for survival, using median survival data for comparison). Values are expressed as mean ± SEM, where $^*p \leq 0.05$, $^{**}p \leq 0.01$, $^{***}p \leq 0.001$, and $^{****}p \leq 0.0001$ are considered significant.

## Reporting summary

Further information on research design is available in the Nature Portfolio Reporting Summary linked to this article.

## Data availability

Genes used for the phylogenetic tree (Figs. S1 and S2) are labeled by their accession number in TRiTRypDB (numbers without asterisk) or NCBI Genbank (numbers with asterisk) and are shown in Table S1. Nanobody sequences have been provided within Fig. S4a and part of the data presented in this work can be accessed in the patent application PCT/EP2022/058575. Source data are provided with this paper.

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

## Acknowledgements

We would like to thank Ella Omasta, Marie-Therese Detobel, Maria Slazak, Yvon Elkrim, Ilham Ahmidi, Ellen Vaneetvelde, Maité Schuurmans, and Nadia Abou for technical and administrative assistance, as well as Prof. Dr. Geert Raes and Prof. Dr. Elias Lolis for the constructive discussions. We also acknowledge Prof. Dr. Peter Takac (Slovak Academy of Sciences) for providing *Glossina morsitans morsitans* pupae. This work was performed in frame of an Interuniversity Attraction Pole Program (PAI-IAP N. P7/41, http://www.belspo.be/belspo/iap/index_en.stm) and was supported by the Strategic Research Program (SRP3, SRP47 and SRP63, VUB), the FWO (FWO G015016N and KAN 1515813 N), the National Institute of Health (NIH) (R21CA259345), "Action de Recherches Concertées" of the University of Brussels (ULB) (ARC ADV), and the Fonds de la Recherche Scientifique (F.R.S.-FNRS) (PDR T.0159.13). BS was supported by the Strategic Research Program (SRP3 and SRP47, VUB). LL was supported by ARC of the University of Brussels (ULB). GC is supported by FWO and the University of Antwerp (www.uantwerpen.be; grant number TT-ZAPBOF 33049) and is member of the Centre of Excellence 'Infla-Med' (www.uantwerpen.be/infla-med) and EU-COST Action OneHealthDrugs (CA21111). HK is supported by internal funding from KU Leuven (C14/23/135). JAVG is supported by the FWO and is a member of the EU-COST action Mye-EUNITER. The funders had no role in study design, data collection and analysis, decision to publish, or preparation of the manuscript.

## Author contributions

B.S. and P.D.B. initiated the study; B.S. and C.D.T organized and designed the study; B.S., I.V.M., L.L., E.H., E.R., C.V., W.B., S.S., H.K., M.W., J.E.P.T., D.P.M., L.B., O.C., M.S.L., M.C., S.H., D.M., K.R., and C.D.T. performed the experiments and analyzed the data; G.H.G provided the Nanobody services; H.R. and I.V.M provided the structural data analysis; G.C. provided the trypanosomatids and ensured the tsetse fly work; B.S., P.D.B., S.M., J.V.G, and C.D.T. wrote the paper; B.S., P.D.B., S.M., and J.V.G. provided the funding for the research.

## Competing interests

B.S., P.D.B., J.V.G., C.V., E.R., S.M., C.D.T., and G.C. are listed as inventors on the patent application PCT/EP2022/058575. The authors declare that the research was conducted in the absence of any commercial or financial relationships that could be construed as a potential conflict of interest. There are no more competing interests.

## Additional information

¹Brussels Center for Immunology, Vrije Universiteit Brussel, Brussels, Belgium. ²Myeloid Cell Immunology Laboratory, VIB Center for Inflammation Research, Brussels, Belgium. ³Structural Biology Brussels, Vrije Universiteit Brussel, Brussels, Belgium. ⁴VIB-VUB Center for Structural Biology, Brussels, Belgium. ⁵Biology of Membrane Transport Laboratory, Université Libre de Bruxelles, Gosselies, Belgium. ⁶Laboratory of Molecular Parasitology, IBMM, Université Libre de Bruxelles, Gosselies, Belgium. ⁷VIB Nanobody Core, Vrije Universiteit Brussel, Brussels, Belgium. ⁸Laboratory of Hepatology, Department of Chronic Diseases and Metabolism, KU Leuven, Leuven, Belgium. ⁹Center for Microscopy and Molecular Imaging (CMMI), Université Libre de Bruxelles, Gosselies, Belgium. ¹⁰Instituto de Investigaciones Biotecnológicas, Universidad Nacional de San Martín-CONICET, Buenos Aires, Argentina. ¹¹Laboratory of Microbiology, Parasitology, and Hygiene (LMPH), Infla-Med Centre of Excellence, University of Antwerp, Antwerp, Belgium. ¹²Amphibian Evolution Lab, Biology Department, Vrije Universiteit Brussel, Brussels, Belgium. ¹³Laboratory of Biomedical Research, Ghent University Global Campus, Incheon, South Korea. ¹⁴These authors contributed equally: Jo A. Van Ginderachter, Carl De Trez. ✉e-mail: benoit.stijlemans@vub.be

