## [Peer Review File · Nature Communications]

Q586B2 is a crucial virulence factor during the early stages of *Trypanosoma brucei* infection that is conserved amongst trypanosomatidsREVIEWER COMMENTS

Reviewer #1 (Remarks to the Author):

The study characterizes Q586B2 role in Trypanosoma infection and as a potential target. The article covers an important, neglected pathogen for which there is an urgent need for new therapeutic strategies. The report is well written with well-performed animal experiments. However, there are a few concerns.

Please find below comments/concerns to consider:

1. In this study the authors suggest Q586B2 as a therapeutic target for Trypanosoma. The genetic knockout failed to demonstrate any survival impact in the fly vector which is reasonable given Q586B2 proposed mechanism of action. However, most animals with neutralizing antibodies /nanobodies that block Q586B2 activity still had 100% mortality by day 30 to 40 (Figure 6 and 7). This raises the question of how effective of a target is Q586B2?
2. It remains unclear how the Q586B2 protein which is located in membrane bound intracellular vesicles is blocked by nanobodies/antibodies.
3. Given that the authors have antibodies against Q586B2, we suggest the gene knockout be demonstrated at the protein level by immunoblot and not only by PCR to show that in these strains protein expression is completely absent.
4. Q586B2 is suggested to be a MIF like protein based mainly on tautomerase activity, but to date, it remains unclear if MIF tautomerase activity is biologically relevant.
5. In Figure 1, the authors suggest structural similarities between Q586B2 and MIF, however MIF forms a trimer while Q586B2 is tetrameric.

Reviewer #2 (Remarks to the Author):

In this manuscript Stijlemans et al describe a T.brucei factor (expressed at all developmental stages and conserved among divergent trypanosomes) that might play a role in dampening infection through interference with the host immune response.

First off, the fact that this factor is expressed at all stages of the paracyte life cycle (and also other trypanosomatids that elicit rather different immune responses in their hosts) would make it unlikely that it is "mandatory for the establishment of infection". It is clearly not (as per the data provided herein).

But perhaps it is something that interferes with infection. The data that infection peaks contain 80% fewer parasites, and that recombinant Q586B2 can induce IL10 production in myeloid cells is clear. So the question now is, what does this actually mean, in terms of the function of this molecule?

The hypothesis proposed here is that the molecule dampens the immune response allowing for proper parasite growth. An equally plausible hypothesis would be that the molecule has little to do with the immune response directly, but instead, serves as a quorum sensing (or rather quorum quenching) factor, so that cells turn into stumpy's a lot faster in vivo (as stumpy's exit cell cycle, this would lead to a drop in apparent parasitemia). PAD1 staining could be used to assess this. Conversely, the authors note that "The number of tsetse flies exhibiting a midgut infection was

similar in both conditions (Fig. 3b)." (line 232). But how about the numbers in the midgut? a quorum sensing/quenching molecule would lead to lower "parasite load" in the fly as well. It is definitely worth noting in this regard that aconitase (an enzyme with tautomerase properties) is key to the differentiation of bloodstream form parasites to procyclics, even in vitro.

Experiments along these lines should be done before seeking an immune mediator phenotype for the current data.

Reviewer #3 (Remarks to the Author):

Review report for NCOMMS-23-14240

In this manuscript, Stijlemans et al uncover the critical role of the newly identified virulence factor Q586B2 in promoting infection by inducing IL-10-mediated responses, facilitating invasiveness. The authors also demonstrate that specific Q586B2 blockade with nanobodies hamper myeloid-derived IL-10 production, reducing parasitaemia. Furthermore, the authors elegantly demonstrate that prophylactic immunisation with Q586B2 protects against subsequent challenges with a wide range of Trypanosomatids, which can be exploited for ongoing intervention strategies. Overall, I find the study compelling and insightful, and I hope my comments help the authors improve their manuscript.

General comments and questions:

1. Is Q56B2 conserved outside of kinetoplasts? Are there homologues (even with low homology) in other organisms including vertebrates?
2. Is it likely for Q56B2 to be exported as a dimer or as larger complexes? Do the authors know whether oligomers/oligomerisation is required to promote cytokine production production?
3. For figure 2b, did the authors used WT parasite strains or parasites expressing the tagged version of the protein?
4. The subcellular distribution pattern of Q586B2 is quite remarkable, but wonder if it colocalises with known vesicular markers such as Rab5/Rab11/p67, or flagellar pocket proteins such as ISG75? The distribution pattern shown in 2b can also be attributed to ER localisation and wonder if a BiP staining has been included to explore this? This is discussed in lines 418-425, but it is worth indicating this here.
5. Regarding the results presented in figure 3a, I wonder if it is normal for the 90:13 strain to tolerate such high densities without triggering stumpy forms?
6. Have the authors measure the cytokine levels of serum and peritoneal lavages from mice infected with WT and KO parasites? Is it possible that the proposed enhanced inflammatory environment in the peritoneal cavity is precluding the myeloid compartment to produce IL-10 production?
7. The levels of Q586B2 in circulation are relatively high according to figure 5C (~0.1 µg/ml). At this concentration, one would expect to see IL-6 and TNF (and probably MIF) as well as IL-10 production by macrophages, according to the results presented in figure 5d. In this context, one could predict a mixed environment in the peritoneum containing both pro- and anti-inflammatory cytokines derived from myeloid cells and the effects are unlikely to be restricted to IL-10 production. Although I agree that Q586B2 has an undeniable effect on IL-10 production, it also seems to trigger other responses. This point could be discussed in a bit more detail in the text. Also, all the stats should be included in the figures to be able to draw conclusions from the data presented (some comparisons are missing).
8. I wonder if the authors have tried recapitulating the results in figure 5d but using peritoneal macrophages instead of bone marrow-derived macrophages?
9. In light of the results presented in figure 4 and 5, I find the results in figure 3C interesting and puzzling. Infection with the KO parasites should in theory result in more initial inflammatory responses, potentially mediated by an exacerbated TNF/IL-6/MIF (and likely IFN γ) production that is not counterbalanced by IL-10. Thus, one would expect more side effects in the tissues as a result of such seemingly exacerbated inflammation. I wonder whether the authors have checked

for signs of pathology during acute and chronic stage of the mice infected with the KO parasites?
10. Along the same lines, if Q856B2 is expressed throughout the course of infection, one would anticipate an effect of Tregs/innate Bregs later on infection. I was wondering if the authors have checked the levels of IL-10 production in these subsets during the "chronic" stage of the infection (e.g., 6dpi vs 20dpi vs 40dpi) with WT and KO parasites?

11. The immunisation experiments are quite compelling but puzzling. It looks as though the immunisation delays the severity of the infection, as demonstrated by the survival curves, but the levels of Tc and Te eventually picked up in circulation. Is this a correct interpretation of the data presented in figure 7A-C? If so, the prophylactic effect seems to be a bit limited, although I agree that treatment has a partial effect, as proposed in line 374. In this context, the term prophylactic seems a bit inappropriate, I think.

Minor comments

12. Line 123, please define pLDDT

13. The statement in line 219 regarding the localisation of Q586B2 to intracellular vesicles can be strengthened if additional vesicular markers are included, as ER retention/expression is also plausible.

14. The survival rate of mice infected with either KO or KD strains are quite different. I wonder if this effect is due to a limited effect of DOX in vivo?

15. The lack of stats in all the relative comparisons in figure 4 makes it a bit hard to interpret the results. For instance, I find it a bit hard to determine whether KO parasites affect the peritoneal PMNs and B cells? Also, if the numbers are so different between strains, would it not be better to reflect these data as frequencies?

16. For the sandwich ELISA presented in figure 5A, would it be helpful to include a negative control to rule out unspecific binding? Although I appreciate this is unlikely, it would be informative to determine what the background signal is for this assay.

17. Some of the statistical parameters are missing in figure 5. For instance, a significant level is shown for MIF at high dose and a "NS" is shown for the lowest dose, but the corresponding analysis is not included for the rest of the doses used in this assay. Please include all the relevant statistical comparisons, irrespective of the level of significance.

18. The levels of MIF production with buffer are quite high. It's worth discussing this.

19. Out of curiosity, is there sequence/structural motifs conserved between Q586B2 and host MIF? Can Nb39 recognise host MIF?

Point-by-point reply to the reviewer's comments:

Reviewer #1 (Remarks to the Author):

The study characterizes Q586B2 role in Trypanosoma infection and as a potential target. The article covers an important, neglected pathogen for which there is an urgent need for new therapeutic strategies. The report is well written with well-performed animal experiments. However, there are a few concerns.

Please find below comments/concerns to consider:

1. In this study the authors suggest Q586B2 as a therapeutic target for Trypanosoma. The genetic knockout failed to demonstrate any survival impact in the fly vector which is reasonable given Q586B2 proposed mechanism of action.

However, most animals with neutralizing antibodies/nanobodies that block Q586B2 activity still had 100% mortality by day 30 to 40 (Figure 6 and 7). This raises the question of how effective of a target is Q586B2?

It is indeed correct that a genetic KO or functional blockade of Q586B2 significantly prolongs the lifespan of the infected animals, but does not provide a cure. To temper the notion that Q586B2 could be considered as a potentially curative therapeutic target, we removed this statement from the title and the abstract (as also requested by the editor) and we rephrased the Discussion section. We also specify that the protective effect of anti-Q586B2 Nanobodies or Q586B2 immunization is a partial protection, meaning that survival is significantly prolonged, but the animals are not cured and ultimately all succumb to the disease.

Concerning the *in vivo* effect of the anti-Q586B2 nanobody treatment, the Discussion (page 21) now reads: "Of note, the effect on survival of using blocking Nanobodies is less pronounced than the effect of knocking-out the gene in *T. brucei*, which could be due to the rapid clearance rate of the Nbs and/or the fact that treatment was halted after 4 days post infection. However, the pharmacokinetics of the Nb constructs could be improved by reformatting them into a half-life extended format or, alternatively, by injecting the Nbs for a longer period of time (post peak parasitemia)."

Concerning the *in vivo* effect of Q586B2 immunization, the Discussion (page 21-22) now reads: "Hence, we considered an immunization with Q586B2, which was indeed found to attenuate peak parasitemia and prolong host survival. It is important to realize that the effect on survival in this setting, though significant, may be hampered by the fact that these parasites are able to escape elimination via migration into tissues (dermis, fat tissue, brain) that are less accessible to antibodies. Nevertheless, its partially protective effect is not only observed in a homologous, but also in a heterologous setting..."

2. It remains unclear how the Q586B2 protein which is located in membrane bound intracellular vesicles is blocked by nanobodies/antibodies.

We provided evidence (shown in Fig 5a-c), using an in house generated Nb-based sandwich ELISA, that this protein is secreted. This fact is the topic of one of the sections in the manuscript and is mentioned in the title of that section ("**Q586B2 is a secreted protein that is able to induce early IL-10 secretion by myeloid cells *in vivo*, promoting *T. b. brucei* infection onset**"). The protein is found in the parasite's secretome and in the serum of infected animals (Fig. 5a-c), hence it is accessible for Nanobodies. The exact mechanism via which Q586B2 is secreted remains unclear so far. In addition, the results of the co-localization experiment (new data shown in Fig 2b) reveal that Q586B2 is present within early and recycling endosomes, further indicating that it can be released.

3. Given that the authors have antibodies against Q586B2, we suggest the gene knockout be demonstrated at the protein level by immunoblot and not only by PCR to show that in these strains protein expression is completely absent.

We tried to perform western blotting using the anti-Q586B2 nanobodies (Nbs) but did not manage to get a signal in WT parasites, most likely because Nbs tend to recognize conformational epitopes that are lost in a SDS page. It is indeed a common feature of Nbs that they are less suitable for western blotting.

However, the KO phenotype at the protein level was confirmed at several levels:

- (i) performing intracellular flow cytometry with anti-Q586B2 Nbs (+ Alexa488-labeled anti-HA) strongly detected the Q586B2 protein in WT parasites, but not in KO parasites (Figure S5a, right panel). As a matter of fact, KO parasites were consistently used as negative control in our flow cytometry experiments.
- (ii) Q586B2 protein is detected via a sandwich ELISA in the secretome of WT *T. b. brucei* parasites, but not in the secretome of KO parasites (Figure 5b).
- (iii) Fig. S5B (formerly Fig. S4c) also demonstrates the absence of the Q586B2 protein, as detected via intracellular flow cytometry, in the dox-inducible Tb927.6.4140 KO strain upon 48h of doxycycline treatment.

Altogether, we feel that these data firmly establish the KO phenotype at the protein level, both in the constitutive and in the dox-inducible KO parasites.

4. Q586B2 is suggested to be a MIF like protein based mainly on tautomerase activity, but to date, it remains unclear if MIF tautomerase activity is biologically relevant.

We picked up the Q586B2-coding gene while searching for genes with MIF-like characteristics, such as an amino terminal proline residue immediately following the initial methionine residue and a maximum size of 115 amino acids, by using the first 25 amino acids of the Q4Q413 protein for blasting. Importantly however, our further analysis of

Q586B2, as described in the first section of the manuscript, soon revealed that there was no homology with MIF, but rather with the *T. cruzi* protein Q4D6Q6 and weakly with the 4-OT tautomerase of the Archaean species *Archaeoglobus fulgidus*. Hence, we do not claim that Q586B2 is a MIF-like protein.

We agree that the physiological relevance of MIF's tautomerase activity is not known. We do not use the (weak) tautomerase activity of Q586B2 as an argument to classify it as a MIF homologue. MIF was merely used as a positive control in the tautomerase assay, as a known protein with tautomerase activity. This is now also clearly mentioned in the revised version of the manuscript (page 8).

5. In Figure 1, the authors suggest structural similarities between Q586B2 and MIF, however MIF forms a trimer while Q586B2 is tetrameric.

In Figure 1, a structural comparison is made between Q586B2 and the *T. cruzi* protein Q4D6Q6. We did not include MIF in this comparison. Q4D6Q6 forms a propeller-like homotetramer, as shown by the solved crystal structure. Q586B2 is predicted by ColabFold to adopt a similar tetrameric structure. In accordance, recombinant Q586B2 purifies as a tetramer (Fig S3a). This is indeed fundamentally different from the trimeric MIF structure, and is in line with our statement that trypanosomes do not harbor a bona fide MIF homologue (which we mentioned on page 6-7). To avoid confusion, we now clearly stated that Q586B2 is not a MIF-like protein and is structurally different from MIF (page 8).

Reviewer #2 (Remarks to the Author):

In this manuscript Stijlemans et al describe a *T. brucei* factor (expressed at all developmental stages and conserved among divergent trypanosomes) that might play a role in dampening infection through interference with the host immune response.

First off, the fact that this factor is expressed at all stages of the parasite life cycle (and also other trypanosomatids that elicit rather different immune responses in their hosts) would make it unlikely that it is "mandatory for the establishment of infection". It is clearly not (as per the data provided herein).

We agree with the reviewer that Q586B2 is not absolutely "mandatory for the establishment of the infection". We feel that our data do, however, convincingly point towards an important regulatory role of Q586B2 in the establishment of *T. b. brucei* and other trypanosomatid infections.

Therefore, we rephrased the abstract, which now reads as: "Collectively, we uncovered a conserved protein playing an important regulatory role in Trypanosomatid infection establishment".

But perhaps it is something that interferes with infection. The data that infection peaks contain 80% fewer parasites, and that recombinant Q586B2 can induce IL10 production in myeloid cells is clear. So the question now is, what does this actually mean, in terms of the function of this molecule?

The hypothesis proposed here is that the molecule dampens the immune response allowing for proper parasite growth. An equally plausible hypothesis would be that the molecule has little to do with the immune response directly, but instead, serves as a quorum sensing (or rather quorum quenching) factor, so that cells turn into stumpys a lot faster *in vivo* (as stumpys exit cell cycle, this would lead to a drop in apparent parasitemia). PAD1 staining could be used to assess this.

Evidence that myeloid cell-derived IL-10 is functionally important in the *in vivo* effect of Q586B2 was provided by the similar first peaks of parasitemia from WT and KO parasites in macrophage/neutrophil-specific IL-10 conditional KO mice (*LysM-cre x Il10fl/fl*). Hence, we go beyond merely showing an induction of IL-10 by Q586B2, we demonstrated an actual *in vivo* importance of this cytokine.

However, the hypothesis of the reviewer that Q586B2 could also serve as a quorum sensing molecule may indeed provide a complementary explanation for the observed reduction in peak parasitemia. Hence, as suggested by the reviewer, we analyzed PAD1 expression and showed that *Tb927.6.4140* KO parasites indeed have a higher PAD1 expression compared to the WT parasites, suggesting a higher propensity to turn into stumpys (new Fig. S5c). These new experimental data are now incorporated in the revised manuscript (results section: page 13 and M&M section: page 32) and the Discussion (page 21) has been changed to acknowledge this point.

Conversely, the authors note that "The number of tsetse flies exhibiting a midgut infection was similar in both conditions (Fig. 3b)." (line 232). But how about the numbers in the midgut? a quorum sensing/quenching molecule would lead to lower "parasite load" in the fly as well. It is definitely worth noting in this regard that aconitase (an enzyme with tautomerase properties) is key to the differentiation of bloodstream form parasites to procyclics, even *in vitro*. Experiments along these lines should be done before seeking an immune mediator phenotype for the current data.

We have determined the parasite load in the fly midgut via PCR and did not observe any significant difference between WT and *Tb927.6.4140* KO parasites, indicating that Q586B2 mainly plays a key role in the mammalian host. These data have now been incorporated in Figure 3b of the revised version. Since there was no difference in infectivity between male and female flies, we now only show data of female flies in Figure 3b.

Reviewer #3 (Remarks to the Author):

Review report for NCOMMS-23-14240

In this manuscript, Stijlemans et al uncover the critical role of the newly identified virulence factor Q586B2 in promoting infection by inducing IL-10-mediated responses, facilitating invasiveness. The authors also demonstrate that specific Q586B2 blockade with nanobodies hamper myeloid-derived IL-10 production, reducing parasitaemia. Furthermore, the authors elegantly demonstrate that prophylactic immunisation with Q586B2 protects against subsequent challenges with a wide range of Trypanosomatids, which can be exploited for ongoing intervention strategies. Overall, I find the study compelling and insightful, and I hope my comments help the authors improve their manuscript.

General comments and questions:

1. Is Q56B2 conserved outside of kinetoplasts? Are there homologues (even with low homology) in other organisms including vertebrates?

We have extended our analysis through BLAST searches. First of all, and surprisingly, this analysis did not reveal any related genes in aquatic *Trypanosoma* species. In addition, no homologous genes were found in any kinetoplastid genome outside Bodonidae and Trypanosomatidae, nor in those of any other eukaryote supergroup. This pattern of occurrence suggests that the gene family to which *Tb927.6.4140* (encoding Q586B2) belongs arose in a direct ancestor of the Bodonidae and Trypanosomatidae sister families. This has now been included in the revised manuscript in the section "**Q586B2 is an evolutionary conserved protein within the related families Trypanosomatidae and Bodonidae**".

2. Is it likely for Q56B2 to be exported as a dimer or as larger complexes? Do the authors know whether oligomers/oligomerisation is required to promote cytokine production?

The ColabFold prediction is that Q586B2 is a homotetramer. Moreover, one of its closest homologues, the *T. cruzi* protein Q4D6Q6, has been crystallized as a homotetramer.

To experimentally address the oligomerization of the Q586B2 protein, we loaded the purified recombinant Q586B2 protein on a size exclusion column (Superdex75 (10/300)) using FPLC chromatography and, based on a standard, we confirmed that the protein predominantly (95%) exists as a tetramer (estimated MW: 55 kDa). Of note, a minor peak of higher molecular weight was also observed, yet this did not induce any cytokine production. To accommodate these findings, we changed Figure S3a by including the FPLC chromatogram. Conversely, we removed the reducing SDS-PAGE, as it does not add any

additional information. Consequently, the figure legend as well as the text describing these data (page 8) have been updated.

3. For figure 2b, did the authors used WT parasite strains or parasites expressing the tagged version of the protein?

We have used the WT parasites and intracellularly stained them with an HA-tagged anti-Q586B2 Nb + Alexa488-anti-HA mAb to visualize the presence of the protein. This has been mentioned in the Figure legend. To avoid confusion, we have now included in all Figure legends the used parasite strain.

4. The subcellular distribution pattern of Q586B2 is quite remarkable, but wonder if it colocalises with known vesicular markers such as Rab5/Rab11/p67, or flagellar pocket proteins such as ISG75? The distribution pattern shown in 2b can also be attributed to ER localisation and wonder if a BiP staining has been included to explore this? This is discussed in lines 418-425, but it is worth indicating this here.

We performed co-localization experiments with known vesicular markers (Rab2b, 5b, 11 and p67) as well as the ER marker BiP. These results, shown in Figure 2b, revealed a partial overlap of Q586B2 protein expression with early endocytic and recycling vesicle markers but no co-localization with p67 nor BiP. In addition, staining with the glycosomal marker Glycerol Kinase showed an overlap with the presence of Q586B2, which is in line with published data (Colasante, C. et al. Comparative proteomics of glycosomes from bloodstream form and procyclic culture form *Trypanosoma brucei brucei*. *Proteomics* **6**, (2006)). Finally, a partial overlap was seen between the presence of Q586B2 and the acidocalcisome marker VIT. Overall, these localization data suggest that the protein can be released, which is in line with the detection of Q586B2 in the secretome of in vitro cultured parasites and in the serum of infected animals (Fig. 5b and c). These new results are now included in the modified Fig.2b and the figure legend, the text (page 9), the M&M section (page 32), and the discussion (page 19-20) have been modified to accommodate these changes.

Of note, the data originally shown in Fig.2a (binding potential of the different Nbs) were now moved to supplemental Figure S4b.

5. Regarding the results presented in figure 3a, I wonder if it is normal for the 90:13 strain to tolerate such high densities without triggering stumpy forms?

The same strain was used in a study on the role of Kinesin heavy chain in trypanosome development (De Muylder G. et al., A *Trypanosoma brucei* kinesin heavy chain promotes parasite growth by triggering host arginase activity. *PLoS Pathog.* 2013 Oct;9(10):e1003731. doi: 10.1371/journal.ppat.1003731), where similarly high parasitemia

levels were recorded. Although short stumpy (SS) forms were not yet studied in this particular strain, it is important to mention that many monomorphic *T. brucei* strains (AnTat1.1A, MiTat1.2, ...) exhibit extremely high parasitemia levels in mice, leading to early uncontrolled parasitemia and premature host death. However, after being passaged into rabbit, they switch to SS forms at a relatively low density.

Of note, since the parasitemia of the 90.13 strain is controlled in mice, the occurrence of SS differentiation is likely. We now provide new data (Figure S5c), based on PAD1 expression, that SS differentiation may be more prominent in the KO strain.

6. Have the authors measured the cytokine levels of serum and peritoneal lavages from mice infected with WT and KO parasites? Is it possible that the proposed enhanced inflammatory environment in the peritoneal cavity is precluding the myeloid compartment to produce IL-10 production?

We have performed a MSD cytokine analysis on peritoneal lavage and serum from WT and KO parasites at 18 hours post injection. Cytokine levels are still very low at this very early time point, except for the inflammation-driving cytokine MIF. MIF has been reported by us as an instrumental cytokine in the induction of inflammatory pathology by African trypanosomes (Stijlemans B. et al. MIF contributes to *Trypanosoma brucei* associated immunopathogenicity development. PLoS Pathog. 2014 Sep 25;10(9):e1004414. doi: 10.1371/journal.ppat.1004414.). Interestingly, MIF induction is significantly higher early after infection (18h) in the peritoneal lavage and serum of KO parasite-infected mice as compared to WT parasite-infected mice. These data have now been included in the revised version of the manuscript in (Fig S8b) and the results section (page 13).

7. The levels of Q586B2 in circulation are relatively high according to figure 5C (~0.1 µg/ml). At this concentration, one would expect to see IL-6 and TNF (and probably MIF) as well as IL-10 production by macrophages, according to the results presented in figure 5d. In this context, one could predict a mixed environment in the peritoneum containing both pro- and anti-inflammatory cytokines derived from myeloid cells and the effects are unlikely to be restricted to IL-10 production. Although I agree that Q586B2 has an undeniable effect on IL-10 production, it also seems to trigger other responses. This point could be discussed in a bit more detail in the text. Also, all the stats should be included in the figures to be able to draw conclusions from the data presented (some comparisons are missing).

It is indeed correct that Q586B2 does not uniquely influence IL-10 production. To better acknowledge this fact, we now wrote in the Results section (page 15):

"A 48h *in vitro* culture of bone marrow-derived macrophages (BMDMs) with different concentrations of endotoxin-free recombinant Q586B2 protein resulted in a prominent induction of IL-10 secretion by even the lowest Q586B2 concentration, while this was not observed for TNF, IL-6 or MIF. TNF and IL-6 are, however, induced by higher Q586B2 concentrations."

Moreover, we included in the Discussion (page 20) a sentence stating "Our *in vitro* data demonstrate that Q586B2 can induce both pro- as well as anti-inflammatory cytokines, yet it exhibits a more potent effect on IL-10 production, at least in macrophages, via an as yet unknown mechanism."

Statistics were included in all figures.

8. I wonder if the authors have tried recapitulating the results in figure 5d but using peritoneal macrophages instead of bone marrow-derived macrophages?

We repeated the experiment using peritoneal macrophages and found a similar trend as for the BMDMs, i.e. an induction of IL-10 at lower Q586B2 concentrations but an induction of all tested cytokines at higher Q586B2 concentrations. We now added these data to the revised manuscript as Fig S9, and included the following sentence to the Results section (page 14): "Noteworthy, similar results were observed using peritoneal macrophages (Fig. S9)".

9. In light of the results presented in figure 4 and 5, I find the results in figure 3C interesting and puzzling. Infection with the KO parasites should in theory result in more initial inflammatory responses, potentially mediated by an exacerbated TNF/IL-6/MIF (and likely IFN γ) production that is not counterbalanced by IL-10. Thus, one would expect more side effects in the tissues as a result of such seemingly exacerbated inflammation. I wonder whether the authors have checked for signs of pathology during acute and chronic stage of the mice infected with the KO parasites?

As an important sign of pathology associated with *T. brucei* infection, we monitored anemia progression throughout the infection. In the acute phase, animals infected with the WT parasites reach lower red blood cell (RBC) levels (i.e. more pronounced anemia), a feature that becomes even more pronounced in the chronic phase (day 22 p.i.).

Also ALT (liver specific injury marker) and AST (more general tissue injury marker) levels were measured at day 6 p.i. (acute phase) and day 22 p.i. (chronic phase) and were found to be lower in the KO parasite-infected group compared to the WT parasite-infected group, again suggesting a lower pathology with the KO parasite.

Hence, infection-associated pathology appears to be lower with the KO parasite. Most probably, the lower parasite levels observed in the *Tb927.6.4140*-deficient parasite-infected mice at the early stages of infection lead consequently to a lower release of parasite-derived pathogenic molecules (i.e. VSG, CpG,...) that in turn results in less pathology during the course of infection.

These data have now been incorporated in the new Fig. S6 and in the results section of the revised manuscript (page 11). Consequently also the M&M section (page 34) has been updated.

10. Along the same lines, if Q856B2 is expressed throughout the course of infection, one would anticipate an effect of Tregs/innate Bregs later on infection. I was wondering if the authors have checked the levels of IL-10 production in these subsets during the “chronic” stage of the infection (e.g., 6dpi vs 20dpi vs 40dpi) with WT and KO parasites?

The doxycycline data point to an important role for Q586B2 during the first three days of infection. Administering doxycycline (and thus knocking down Q586B2 expression) at day 3 post-infection does no longer affect the first peak of parasitemia and survival. Hence, it seems that the crucial role of Q586B2 is mainly observed in a rather narrow time window, i.e. the very early phase of infection.

Moreover, previously published data from our lab and from several of our co-authors illustrated that an infection with WT *T. brucei* parasites was not affected by regulatory T cells (except upon experimental intervention) (Guilliams M, et al. Experimental expansion of the regulatory T cell population increases resistance to African trypanosomiasis. *J Infect Dis.* 2008 Sep 1;198(5):781-91. doi: 10.1086/590439) and that B-cells can indeed produce IL-10, but without effect on pathology development (Magez S, et al. The role of B-cells and IgM antibodies in parasitemia, anemia, and VSG switching in *Trypanosoma brucei*-infected mice. *PLoS Pathog.* 2008 Aug 8;4(8):e1000122. doi: 10.1371/journal.ppat.1000122.).

For all these reasons, we did not further investigate the role of Treg or IL-10-producing B cells during infections with KO parasites.

11. The immunisation experiments are quite compelling but puzzling. It looks as though the immunisation delays the severity of the infection, as demonstrated by the survival curves, but the levels of Tc and Te eventually picked up in circulation. Is this a correct interpretation of the data presented in figure 7A-C? If so, the prophylactic effect seems to be a bit limited, although I agree that treatment has a partial effect, as proposed in line 374. In this context, the term prophylactic seems a bit inappropriate, I think.

The reviewer is correct. Hence, we omitted the term “prophylactic” throughout the manuscript.

Minor comments

12. Line 123, please define pLDDT

This has been defined in the text (Page 5) and in the legend of Fig. 1.

13. The statement in line 219 regarding the localisation of Q586B2 to intracellular vesicles can be strengthened if additional vesicular markers are included, as ER retention/expression is also plausible.

This question has been addressed (see answer to major point 4 by this reviewer).

14. The survival rate of mice infected with either KO or KD strains are quite different. I wonder if this effect is due to a limited effect of DOX in vivo?

Different mechanisms may account for these results. Two plausible hypotheses are: (i) DOX is added in the drinking water, but at peak parasitemia or during the later stages of infection mice typically tend to drink and eat less. Hence, the uptake of DOX during those infection phases may not be sufficient to allow the generation of a full KD phenotype; (ii) parasites tend to migrate rapidly towards the brain or fat tissue and it is unknown to what extent DOX reaches these organs. Hence, parasites residing in these sites may not be fully KD.

15. The lack of stats in all the relative comparisons in figure 4 makes it a bit hard to interpret the results. For instance, I find it a bit hard to determine whether KO parasites affect the peritoneal PMNs and B cells? Also, if the numbers are so different between strains, would it not be better to reflect these data as frequencies?

When no statistics are added, this means that the relative comparison is not statistically significant. To comply with the reviewer's remark, we included 'ns' to the figures when there is no significant difference. However, this would interfere with the clarity of some figures, in which case it was mentioned in the text.

In the peritoneum of *Tb927.6.4140* KO-infected mice, a trend (not statistically significant) towards more B cells and less PMN is observed as compared to WT-infected mice.

The frequencies of peritoneal CD45+ immune cells in WT- versus *Tb927.6.4140* KO-infected mice are shown in the pie charts of Fig 4a and are plotted for each CD45+ immune cell population separately in Figure S7. When using frequencies, a rise in the % of one population will always result in a % decrease of other populations (since the sum remains 100%), while the absolute number of the latter populations may not change or even increase. Hence, frequencies (%) and absolute numbers provide complementary information, but we feel that absolute numbers better reflect the actual composition of the peritoneal immune compartment, which is why we included those data in the main Figure 4b.

16. For the sandwich ELISA presented in figure 5A, would it be helpful to include a negative control to rule out unspecific binding? Although I appreciate this is unlikely, it would be informative to determine what the background signal is for this assay.

In the results shown in Figure 5a, background levels were subtracted prior to calculating the concentration of Q586B2. This has now been clearly stated in the M&M section (Nb-

based ELISA at page 34). Of note, values of the background signal never reached OD_{450nm} above 0.25.

17. Some of the statistical parameters are missing in figure 5. For instance, a significant level is shown for MIF at high dose and a "NS" is shown for the lowest dose, but the corresponding analysis is not included for the rest of the doses used in this assay. Please include all the relevant statistical comparisons, irrespective of the level of significance.

When no statistics is added, this means that the relative comparison is not statistically significant. To comply with the reviewer's remark we included all the relevant statistical comparisons and 'NS' to the figure when there is no significant difference.

18. The levels of MIF production with buffer are quite high. It's worth discussing this.

Indeed, MIF production by BMDMs stimulated with buffer alone is high, yet similar to unstimulated BMDMs (data not shown). It is shown by others that BMDMs produce nanogram amounts of MIF upon in vitro culturing (Lee SH, et al. Developmental endothelial locus-1 inhibits MIF production through suppression of NF-κB in macrophages. *Int J Mol Med*. 2014 Apr;33(4):919-24. doi: 10.3892/ijmm.2014.1645). This has now also been mentioned in the Results section (page 14).

19. Out of curiosity, is there sequence/structural motifs conserved between Q586B2 and host MIF? Can Nb39 recognise host MIF?

We mentioned in the manuscript that there is little sequence similarity between Q586B2 and MIF. Moreover, we mentioned in the manuscript: "In accordance, the Protein Homology/analogy Recognition Engine Version 2.0 (Phyre2) (<http://www.sbg.bio.ic.ac.uk/phyre2/html/page.cgi?id=index>), that also analyzes structural homologies between proteins, yielded no homology between Q586B2 and MIF."

Together, these data demonstrate little sequence and structural similarity between Q586B2 and MIF. Consequently, Nb39 did not bind to mouse MIF.

REVIEWERS' COMMENTS

Reviewer #3 (Remarks to the Author):

I am satisfied with the revisions and thus recommend the current version for publication. I would like to personally congratulate the authors for addressing all the points and for going above and beyond. I thoroughly enjoyed reading the rebuttal and the new version of the manuscript.